# Remodeling of dermal adipose tissue alleviates cutaneous toxicity induced by anti-EGFR therapy

**Leying Chen[1†], Qing You[1†], Min Liu[1], Shuaihu Li[1], Zhaoyu Wu[1], Jiajun Hu[1], Yurui Ma[1], Liangyong Xia[1], Ying Zhou[1], Nan Xu[2]\*, Shiyi Zhang[1]\***

[1]School of Biomedical Engineering, Shanghai Jiao Tong University, Shanghai, China; [2]Department of Dermatology, Shanghai East Hospital, Tongji University, Shanghai, China

**Abstract** Anti-epidermal growth factor receptor (EGFR) therapy–associated cutaneous toxicity is a syndrome characterized by papulopustular rash, local inflammation, folliculitis, and microbial infection, resulting in a decrease in quality of life and dose interruption. However, no effective clinical intervention is available for this adverse effect. Here, we report the atrophy of dermal white adipose tissue (dWAT), a highly plastic adipose tissue with various skin-specific functions, correlates with rash occurrence and exacerbation in a murine model of EGFR inhibitor-induced rash. The reduction in dWAT is due to the inhibition of adipogenic differentiation by defects in peroxisome proliferator-activated receptor γ (PPARγ) signaling, and increased lipolysis by the induced expression of the lipolytic cytokine IL6. The activation of PPARγ by rosiglitazone maintains adipogenic differentiation and represses the transcription of IL6, eventually improving skin functions and ameliorating the severity of rash without altering the antitumor effects. Thus, activation of PPARγ represents a promising approach to ameliorate cutaneous toxicity in patients with cancer who receive anti-EGFR therapy.

**\*For correspondence:**
xnhrb@aliyun.com (NX);
zhangshiyi@sjtu.edu.cn (SZ)

†These authors contributed equally to this work

## Editor's evaluation

This paper will be of interest to oncologists and dermatologists, and has high clinical relevance. It reveals a novel mechanism of EGFR inhibitor-induced rash which be may closely related to atrophy of dermal white adipose tissue (dWAT). A series of experimental manipulations dissect the mechanism with a murine model, supporting the major claims of the paper.

## Introduction

Epidermal growth factor receptor (EGFR) is a canonical therapeutic target for non-small cell lung cancer, breast cancer, and colorectal cancer. Cutaneous toxicity induced by EGFR inhibitors (EGFRIs) is an inflammatory disorder characterized by papulopustular rash, folliculitis, microbial infection and pruritus. Most commonly, rash is observed in 70–90% of patients receiving anti-EGFR therapy (*Cappuzzo et al., 2010*; *Wu et al., 2017*; *Sequist et al., 2013*), resulting in a decrease in quality of life. Except for the empiric use of corticosteroids and antibiotics, no effective clinical treatment is currently available for these adverse effects, which is responsible for 8–17% of dose modifications and interruptions (*Lacouture, 2006*). Therefore, new therapeutic targets for EGFRI-specific skin toxicity urgently need to be discovered.

Dysfunction of the epidermal barrier, retardation of hair follicle growth and various dermal inflammatory responses have been proposed as causative factors for EGFRI-induced rash, but therapeutic agents that interfere with these mechanisms have limited effectiveness (*Lichtenberger et al., 2013*;

*Mascia et al., 2013*). An analysis of biopsies from rodents with the genetic ablation of EGFR or pharmacological EGFR inhibitors both showed a dramatic loss of dermal adipose tissue (*Murillas et al., 1995*; *Sibilia and Wagner, 1995*; *Sugawara et al., 2010*; *Surguladze et al., 2009*), a newly recognized adipose layer termed dermal white adipose tissue (dWAT). Compared to other conventional fat depots, dWAT has more skin-specific functions and a high plasticity to expand or contract in response to different stimuli (*Chen et al., 2019*). For example, local infection (*Zhang et al., 2015*), a high-fat diet (HFD) and peroxisome proliferator-activated receptor γ (PPARγ) agonists (*Zhang et al., 2019a*) induce dWAT expansion. Recent studies have revealed unique roles for dWAT in skin fibrosis and scarring (*Marangoni et al., 2015*; *Varga and Marangoni, 2017*; *Zhang et al., 2019b*), wound healing (*Plikus et al., 2017*; *Schmidt and Horsley, 2013*; *Shook et al., 2020*), immune modulation (*Dokoshi et al., 2018*; *Schmid et al., 2017*; *Zhang et al., 2015*), and regeneration of hair follicles (*Donati et al., 2014*; *Festa et al., 2011*). During the hair cycle, dermal adipose tissue expands in the anagen phase and regresses in catagen phase, providing vital signals to regulate hair growth (*Zhang et al., 2019b*). In addition, the reduction in dWAT in transgenic and pharmacologically treated mice results in a scaly skin phenotype with delayed hair coat formation, poorly developed pilosebaceous structures and epidermal barrier function (*Chen et al., 2002*; *Herrmann et al., 2003*), and these phenomena resembled the abnormalities in skin observed upon EGFR inhibition. A similar synchronized pattern was also observed in wound healing. Dermal adipocytes surrounding the site of skin injury undergo lipolysis and dedifferentiation, transition to myofibroblasts and attract macrophages to stimulate re-epithelialization and revascularization (*Shook et al., 2020*). Additionally, new adipocytes regenerating from myofibroblasts were also observed in healed wounds (*Plikus et al., 2017*). The importance of dWAT in host defense is the promotion of cathelicidin antimicrobial peptide (Camp) release during adipogenic differentiation (*Zhang et al., 2015*). The production of Camp defends against *Staphylococcus aureus* (*S. aureus*) infection, which often develops along with EGFRI skin toxicities (*Lichtenberger et al., 2013*).

Since the multiple functions of dWAT are closely related to EGFRI-induced skin toxicities and a reduction in dermal fat has been observed in EGFR-depleted mice, we investigated whether dWAT plays a central role in EGFRI-induced rash. Here, using an EGFRI-induced rash model and dermal fibroblast (dFB) differentiation assays, we showed that atrophy of dWAT occurs during rash development and has a strong causality with rash severity. The differentiation of dFBs was obviously inhibited and lipolysis of mature adipocytes was increased upon EGFR inhibition. Notably, stimulating dermal adipocyte expansion with a high-fat diet (HFD) or the pharmacological PPARγ agonist rosiglitazone (Rosi) ameliorated the severity of rash.

## Results
### A reduction in dWAT is a hallmark of the pathophysiology of rash

EGFRI therapy in patients is consistently associated with specific skin toxicities, such as papulopustular rash. Additionally, weight loss and decreased appetite accompany these changes (*Supplementary file 1*). Recent reports have shown a strong, positive correlation between body weight and dWAT thickness (*Kasza et al., 2016*). To investigate whether dWAT responses to EGFR inhibition, we established a rat rash model by repeated oral gavage of Afatinib (Afa), an EGFR inhibitor with a high occurrence of skin toxicity (*Sequist et al., 2013*). Rats continued losing weight and exhibited reduced food intake throughout the Afa intervention, and at the end of the treatment, they had lost up to 27.9% and 23.1% of their initial body weight and food intake, respectively (*Figure 1—figure supplement 1A, B*). After 3 days of treatment, few rats began to occur the skin lesions, until day 7, almost all rats developed rash with 23.08% of them were grade two and above. Grade three rash happened at day eight and increased in the next two days (*Figure 1—figure supplement 1C*). The rat model recapitulated all skin features of patients receiving EGFRIs, including erythema, pustules and itching, and the pustules conflated to lakes of pus that dried and formed yellow crusts when rash developed to a higher grade (*Figure 1A*). Administrations by other EGFRIs, such as Erlotinib, Gefitinib, Osimertinib, had similar phenotypes (*Figure 1—figure supplement 1D*). Immunohistochemical analysis revealed a dense inflammatory infiltrate with a marked clustering of CD68[+] macrophages, CD3[+] T cells, CD11b[+] immune cells, and mast cells in the dermis (*Figure 1—figure supplement 1E,F*). The cytokine expression file showed a similar tendency with human biopsies and mouse

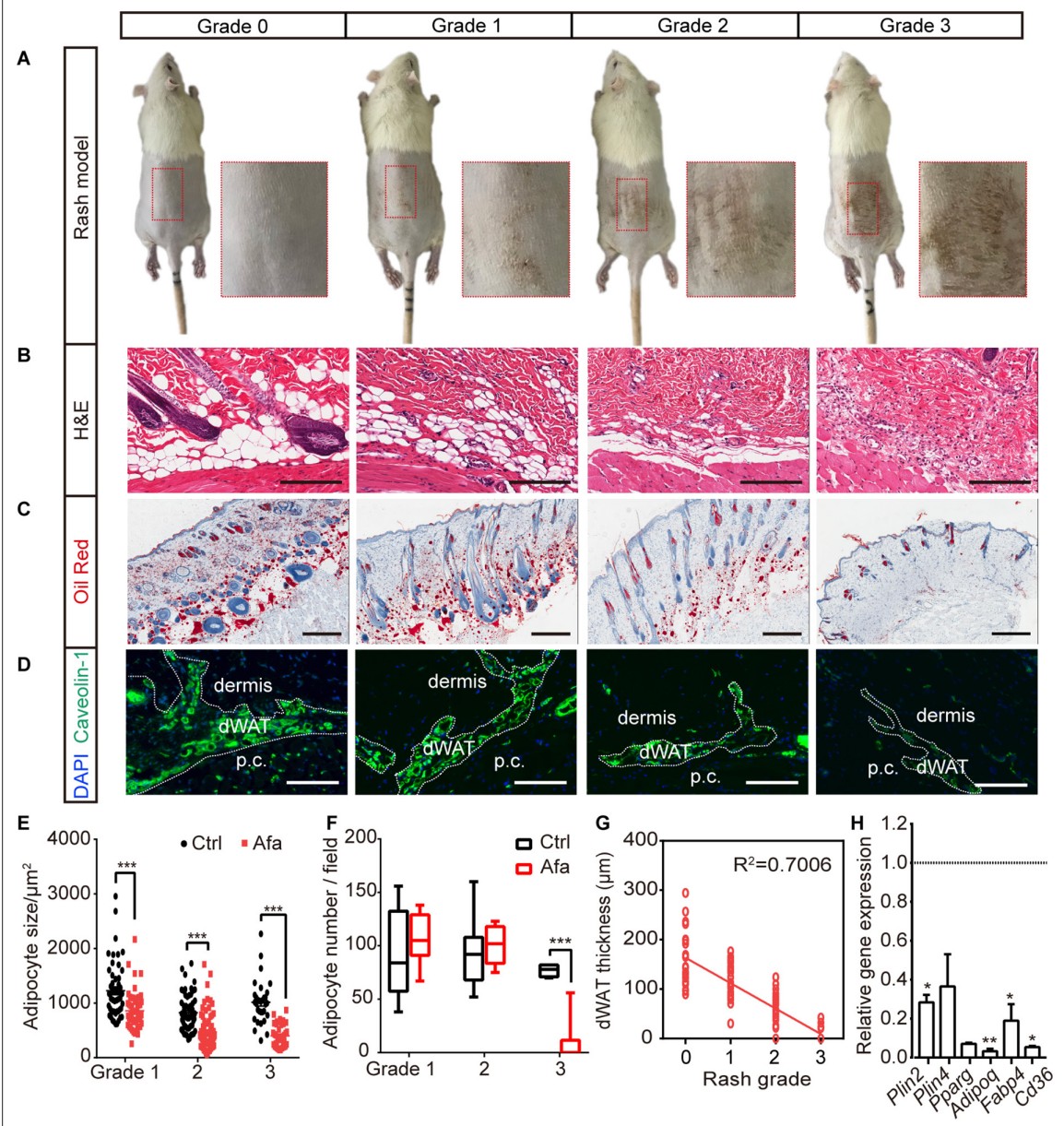

**Figure 1.** The response of dWAT to EGFR inhibition. (**A**) Representative images of the rat rash model. (**B–D**) H&E (**B**), Oil Red (**C**), and Caveolin-1 (**D**) staining of skin from control and Afa-treated rats. Scale bars: 200 μm, 500 μm and 130 μm (top to bottom). p.c. refers to panniculus carnosus. (**E and F**) Quantification of the dWAT size (**E**), and adipocyte number (**F**) in rats with different rash grades compared with the control (Ctrl). n = 3–5 per group. (**G**) Relationship between dWAT thickness and rash grade. The square values of the Pearson's correlation coefficients are shown. (**H**) RT-PCR measurements of mRNA levels of adipogenic genes in skin tissues from control and Afa-treated rats. n = 4–5 per group. Data are presented as the means ± SEM. *p < 0.05, **p < 0.01, and ***p < 0.001 using two-tailed unpaired Student's t test.

The online version of this article includes the following figure supplement(s) for figure 1:

**Figure supplement 1.** Characteristics of rat rash model.

**Figure supplement 2.** Shaving effects rash occurrence and progress.

**Figure supplement 3.** Characteristics of sWAT during EGFR inhibition.

**Figure supplement 4.** Reduced expression of PPARγ and Perilinpin-2 after Afa treatment.

models (*Lichtenberger et al., 2013*; *Mascia et al., 2013*), Afa dramatically increased the expression of proinflammatory chemokines such as *Ccl2*, *Ccl5*, *Cxcl1*, *Cxcl2*, and *Cxcl3* (*Figure 1—figure supplement 1G*).

To analyze the response of dWAT during rash process, skin biopsies were obtained according to the rash process at different grades (*Figure 1—figure supplement 1H*). The thickness of the dWAT layer decreased with an increase in the rash grade, and skin lipid levels also decreased (*Figure 1B and C*). A significant decrease in the number of cells stained with the adipocyte surface protein Caveolin-1 was also observed (*Figure 1D*). A recent study described the importance of hair eruption on rash occurrence (*Klufa et al., 2019*). In our rat model, there was a similar situation that the skin rash occurred earlier and was more severe on shaved back skin than on the unshaved area (*Figure 1—figure supplement 2A, B*). This result indicates that shaving-induced hair eruption has an important impact on rash progress during the EGFR inhibitor treatment. The unshaved area still developed abnormal erythema (*Figure 1—figure supplement 2A*), and showed dWAT reduction after long-term Afa treatment (*Figure 1—figure supplement 2C, D*).

Changes in dWAT were examined by performing a detailed morphometric analysis. The attenuation of intradermal adipocytes was due to a decrease in both size and number, and time-course studies revealed that the size change occurred prior to the change in adipocyte number (*Figure 1E, F*). Consistent with our hypothesis, a positive correlation existed between dWAT thickness and rash grade throughout Afa treatment (*Figure 1G*). This correlation suggests that changes in the dWAT layer in response to a process associated with EGFRI affected skin function. In addition to the dermal adipose layer, skin-associated adipose tissue also includes back subcutaneous adipocytes (sWAT), inguinal adipocytes (iWAT) and brown adipocytes (BAT), which are all adjacent to the back skin. Although clear evidence of anatomically and functionally distinct skin adipose depots in rodents has been reported, for example, the dWAT layer was separated by a distinct muscle layer known as the panniculus carnosus and verified to develop independently of sWAT, these two layers are not quite physically and functionally demarcated in humans (*Chen et al., 2019*; *Wojciechowicz et al., 2013*). Thus, we also monitored the weight change of sWAT and other skin adipose depots, and found that iWAT and BAT also showed a decreasing trend at late time points (*Figure 1—figure supplement 3A*), whereas sWAT represented a more similar response to dWAT during rash occurrence (*Figure 1— figure supplement 3B–D*).

The reduction in the size and number of adipocytes in the dWAT layer at the rash site suggested that adipocytes may be modulated by the inhibition of differentiation and an increase in lipolysis. To examine whether the differentiation of dermal adipocyte was affected, we first compared the expression of differentiation-associated genes in Afa-treated skin with that in vehicle-treated skin using quantitative PCR. The expressions of general adipocyte marker genes were significantly decreased at the end of the treatment (*Figure 1H*). Reduced expression of PPARγ and Perilinpin-2 were also detected in skin samples from rats (*Figure 1—figure supplement 4A, B*). Taken together, these data suggest that the loss of dermal adipocytes is a characteristic hallmark of EGFRI-induced skin toxicity, and this process might be modulated by PPAR-mediated differentiation.

## EGFRI stimulation induces the dedifferentiation of dWAT

A loss of dWAT was also reported to be a prominent feature of systemic sclerosis in humans, a disease with disturbed skin homeostasis and dermal fibrosis. The adipocyte–myofibroblast transition is the primary event in the pathogenesis of cutaneous fibrosis (*Varga and Marangoni, 2017*). Additionally, dermal adipocyte dedifferentiation and subsequent conversion to myofibroblasts have been verified in skin injury and are necessary for efficient tissue repair (*Shook et al., 2020*). To characterize dermal adipocyte differentiation, we first performed Masson's trichrome staining for collagen to characterize dermal adipocyte differentiation. Rats receiving daily Afa treatments showed a time-dependent progressive decrease in dWAT thickness associated with a slightly thicker dermis and collagen deposition in the early stage of rash (*Figure 2A and B*). We next analyzed the expression of fibrotic genes in dWAT before and after rash to evaluate the adipocyte to fibroblast transdifferentiation process. However, dermal adipocytes isolated from the back skin of Afa-treated rats only exhibited slight fibrotic potential in the early stage that diminished quickly (*Figure 2C*), whereas the expression of general adipocyte markers represented a continuous and obvious decrease (*Figure 2D*). The same trend was also observed in sWAT (*Figure 2—figure supplement 1A, B*). These data suggest

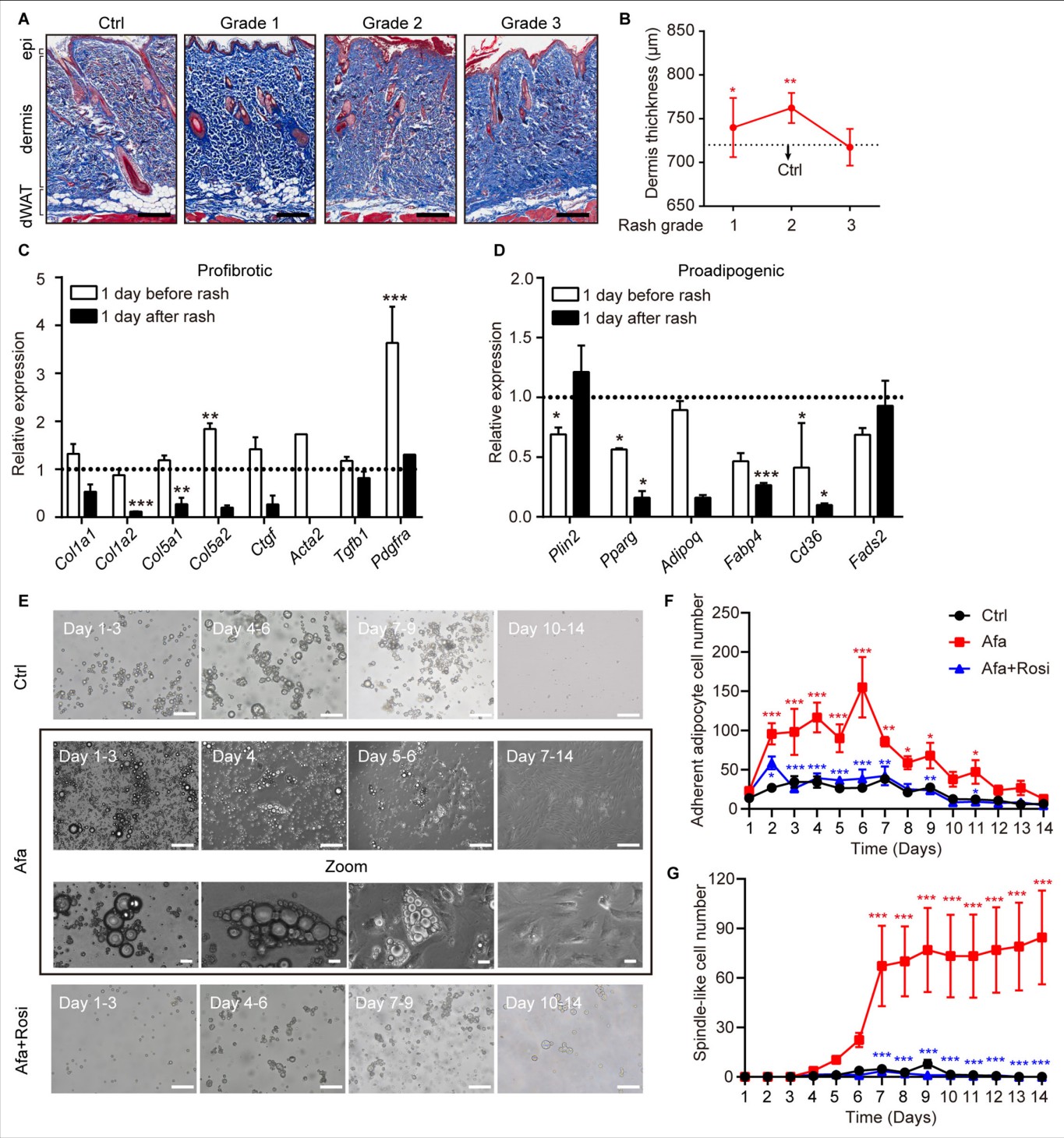

**Figure 2.** Dedifferentiation of mature dermal adipocytes upon EGFR inhibition. (**A**) Masson's trichrome staining of skin sections from rats at the indicated times. n = 3–5 per group. epi refers to epidermis. Scale bars: 500 µm. (**B**) Changes in the thickness of the dermis in Ctrl- and Afa-treated rats. n = 3–5 per group. (**C and D**) RT-PCR measurements of profibrotic genes (**C**), and proadipogenic genes (**D**) in isolated dermal adipocytes at the indicated times, n = 3–4 per group. Data was normalized to Ctrl. Dotted line represented the mean gene expression of the Ctrl group. (**E**) Morphological changes at different time points of Afa (100 nM) or Rosi (5 µM) treatment revealed a transition from mature dermal adipocytes to dedifferentiated adipocytes. Scale bars: 100 µm and 50 µm for zoom pictures. (**F**) Quantification of adherent adipocyte cells. (**G**) Quantification of dedifferentiated, spindle-like cells per field. Data are presented as the means ± SEM (Afa/Rosi group *vs* Ctrl group and Afa +Rosi group *vs* Afa group). *p < 0.05, **p < 0.01, ***p < 0.001 using two-tailed unpaired Student's t test and one-way ANOVA.

The online version of this article includes the following figure supplement(s) for figure 2:

*Figure 2 continued on next page*

*Figure 2 continued*

**Figure supplement 1.** Additional dedifferentiation changes of dWAT and sWAT.

that collagen deposition and myofibroblast transition occur in the early phases of EGFRI-related skin changes and are sustained for only a short period.

To determine whether dedifferentiation exists, ceiling culture was performed using dWAT cells isolated from normal rat back skin. Upon EGFRI stimulation, we found that attached adipocytes began to decrease the size of lipid droplets and the cells formed spindle-like shapes on day four after Afa treatment. Then, they gradually elongated, lost intracellular lipids and proliferated as fully dedifferentiated PDGFRα + fibrocytes (*Figure 2E*; *Figure 2—figure supplement 1C*). In comparison, adipocytes cultured under control experimental conditions attached much fewer and still retained lipid droplets (*Figure 2E–G*). Based on our evidence, the expression of genes related to the function of mature adipocytes, such as *Pparg*, *Adipoq*, and *Fabp4*, was downregulated after EGFRI treatment (*Figure 1H*; *Figure 2D*). The gene enrichment analysis highlighted the significance of PPAR signaling in the pathology of EGFRI-induced skin toxicity. The blockade of PPARγ produces defective adipocytes and exerts a negative effect on skin wound healing (*Schmidt and Horsley, 2013*). In contrast, rosiglitazone (Rosi), a PPARγ agonist, abrogates bleomycin-induced scleroderma and blocks profibrotic responses (*Bi et al., 2016*; *Wu et al., 2009*). Thus, we hypothesized that PPARγ activation might reverse adipocyte dedifferentiation in response to EGFR inhibition through a compensatory mechanism. We evaluated the effect of Rosi on Afa-treated dermal adipocytes to directly test this hypothesis. Application of Rosi reversed the dedifferentiation of dermal adipocytes induced by Afa, and cells cultured in the presence of Rosi displayed little attachment until 14 days (*Figure 2E–G*). Correspondingly, the transcriptional level of *Pparg* was downregulated by Afa and upregulated by Rosi (*Figure 2—figure supplement 1D*). In summary, these results suggest that an EGFR signaling deficiency, caused by treatment with EGFRI, triggers the dermal adipocyte-myofibroblast transition at an early stage and significant dedifferentiation, and the PPARγ agonist Rosi effectively rescues this phenomenon.

## Differentiation of adipocyte precursors are suppressed during EGFR inhibition

Dermal adipocyte precursor cells (APs) were recently identified in the skin *Festa et al., 2011*; moreover, recent genetic and molecular research revealed that the activation of APs depends on the PDGFRA-PI3K-AKT pathway (*Rivera-Gonzalez et al., 2016*). The AKT pathway is a key downstream target of EGFR kinase, which also modulates proliferation and survival. To determine whether the loss of mature dWAT cells was caused by EGFR inhibition on APs, we isolated dFBs from the skin dermis and treated them with Afa for 3, 6 and 9 days during adipogenic differentiation in vitro. Consistent with our hypothesis, the differentiation of dFBs in the presence of Afa was inhibited (*Figure 3A and B*), and the concentration of Afa used in this experiment had little antiproliferative effect (*Figure 3—figure supplement 1A*). Notably, the addition of Rosi circumvented the inhibition of adipogenic differentiation (*Figure 3A*). Next, we aimed to define the molecular pathways affected by EGFR inhibition in dermal adipogenic cells. The gene signature induced by Afa in APs (*Figure 3C*) was similar to the changes in gene expression observed in mature adipocytes (*Figure 2D*). Adipocytes have been shown to express a variety of pattern recognition receptors and cytokine receptors, including TLR1-9, NOD1-2, TNFR, IL4R, and IL6R (*Chen et al., 2019*; *Deiuliis et al., 2011*; *Sindhu et al., 2015*). Interestingly, the expression levels of many of these receptors were increased during Afa treatment (*Figure 3—figure supplement 1B*), indicating the proinflammatory potential of adipocytes upon EGFR inhibition. Additionally, we analyzed the levels of PDGFRα and phosphorylated AKT. Significantly increased levels of PDGFRα were detected on days 6 and 9, indicating a retardation of AP differentiation. Phospho-AKT1 and phospho-AKT2 levels were also affected by Afa (*Figure 3D and E*). Cotreatment with the PPARγ agonist Rosi and the EGFR inhibitor Afa completely abrogated the inhibition of APs differentiation. Notably, the addition of Rosi significantly activated both AKT1 and AKT2 (*Figure 3D and E*). We next examined the number of APs in EGFRI-treated rats. Using FACS, we found that APs (CD31[-], CD45[-], CD34[+], CD29[+]) were absent after EGFR inhibition (*Figure 3F and G*). Together, these data indicate that the loss of mature dermal adipocytes induced by an EGFRI correlates with loss of APs and inhibition of their activation.

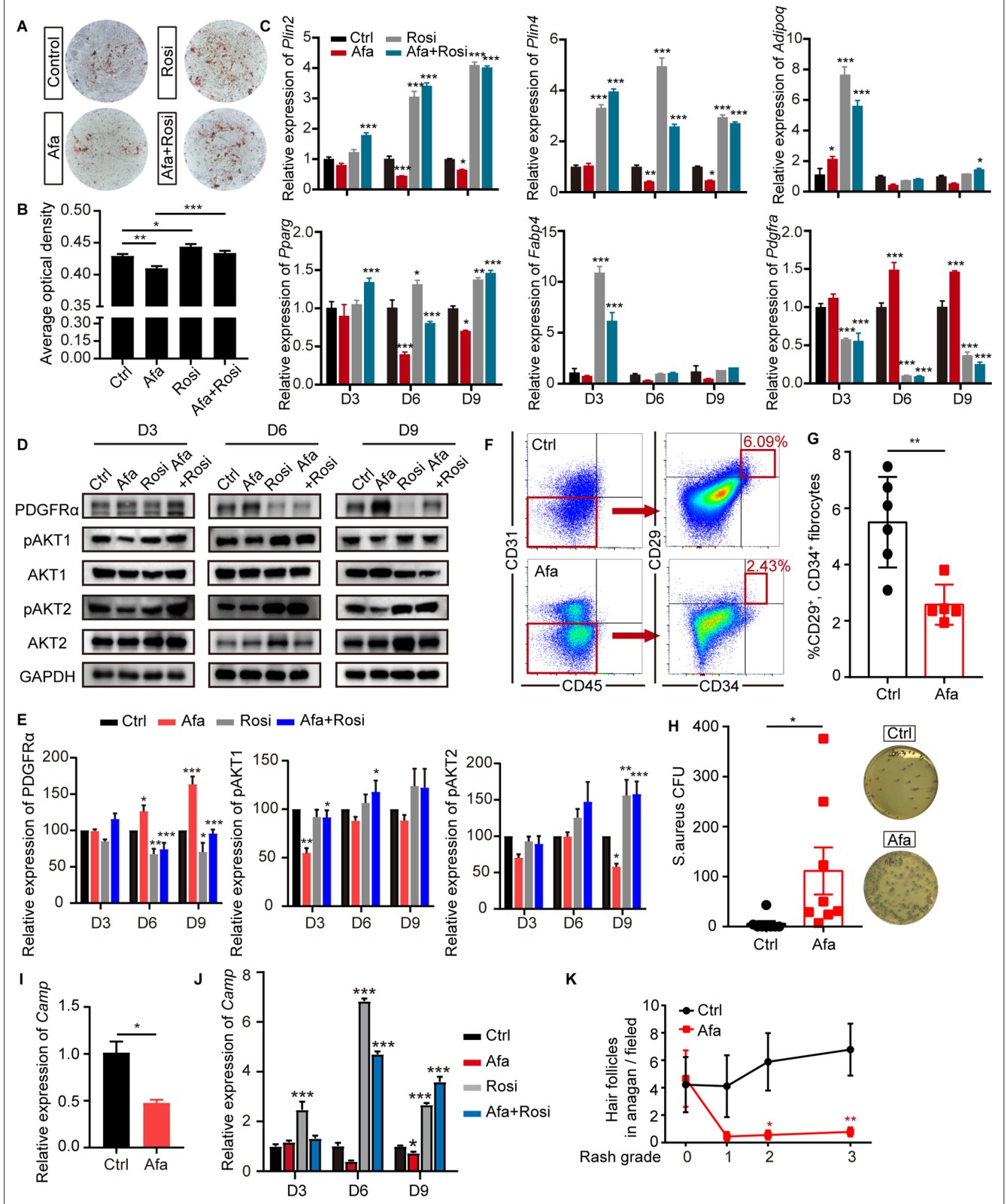

**Figure 3.** EGFR inhibition blocks adipocyte progenitor differentiation. (**A**) dFB cells were isolated from P1 rat skin and then cultured with adipocyte differentiation medium for 9 days in the presence or absence of Afa (10 nM) or Rosi (5 μM). Lipid production was detected using Oil Red staining. (**B**) Quantification of Oil Red staining. (**C**) Relative mRNA expression of adipogenic genes at the indicated time points during differentiation. n = 3 per group. (**D**) Western blot analysis of differentiating dFB cells. (**E**) Quantification of expression level of PDGFRα, pAKT1 and pAKT2 relative to Ctrl.

*Figure 3 continued on next page*

*Figure 3 continued*

(**F**) FACS analysis of adipocyte progenitors in dFB from Ctrl- and Afa-treated rats. (**G**) The percentage of CD31⁻, CD45⁻, CD34⁺, and CD29⁺ fibrocytes was quantified. n = 5–6 per group. (**H**) *S. aureus* growth in media containing skin homogenates from Ctrl- or Afa-treated rats. n = 8 per group. (**I**) Relative mRNA expression of *Camp* from isolated-dWAT at Day 3. n = 5 per group. (**J**) Relative mRNA expression of *Camp* at the indicated time points during dFB differentiation. n = 5 per group. (**K**) Numbers of hair follicles in anagen phase during Afa treatment. n = 3 per group. Data are presented as the means ± SEM. *p < 0.05, **p < 0.01, and ***p < 0.001 (Afa/Rosi group *vs* Ctrl group and Afa +Rosi group *vs* Afa group) using two-tailed unpaired Student's t test and one-way ANOVA.

The online version of this article includes the following source data and figure supplement(s) for figure 3:

**Source data 1.** Original western blot data for *Figure 3D*.

**Figure supplement 1.** Expression levels of inflammatory receptors during dFB differentiation.

Since recent reports have shown that dermal adipocytes have the capacity to inhibit invasive bacterial skin infection and support hair regeneration (*Festa et al., 2011*; ), we determined whether the reduction and inhibition of APs contribute to the loss of these functions. As expected, treatment of Afa resulted in increased *S. aureus* infection (*Figure 3H*). To verify the relationship between Afa-induced dWAT reduction and *S. aureus* infection, we performed transcriptional experiments of dermal adipocyte cells. Correspondingly, the transcriptional level of *Camp* both in isolated dWAT (*Figure 3I*) and differentiated dFB (*Figure 3J*) were decreased by Afa, and Rosi significantly upregulated the expression level (*Figure 3J*). In addition, the number of hair follicles in anagen phase decreased with an increasing rash grade (*Figure 3K*), and rats treated with the EGFRI showed hair follicles with abnormal growth (*Figure 1B*). The levels of phospho-EGFR and ERK in dWAT and the outer sheath of hair follicles were significantly decreased (*Figure 3—figure supplement 1C*). Consequently, the number of proliferative cells characterized by Ki67 immunostaining was reduced, and the number of apoptotic cells detected using TUNEL immunofluorescence staining was increased (*Figure 3—figure supplement 1D, E*). Therefore, our results showed that a lack of APs and mature dWAT might contribute to impairing the maintenance of the host defense and hair growth in the skin.

## EGFR inhibition induces lipolysis through epidermal keratinocytes

In addition to differentiation, the reduction in dermal adipocyte size also suggested that adipocytes might alter their lipid content through lipolysis. To examine whether lipolysis occurs after EGFR inhibition, isolated dermal adipocytes were assessed using transmission electron microscopy, and the expression of lipolysis-activated markers, the rate-limiting enzyme ATGL and phosphorylated hormone-sensitive lipase (pHSL) were detected. We observed increased levels of ATGL and pHSL in dWAT (*Figure 4A and B*). Consistent with these changes, multiple smaller droplets adjacent to large lipid droplets were detected in Afa-treated rats, whereas only large, unilocular lipids were retained in the control group (*Figure 4C*). However, in sWAT, lipolysis-induced glycerol production was not increase in either sWAT isolated from Afa-treated rats or Afa stimulation of sWAT from Ctrl-treated rats (*Figure 4—figure supplement 1A*). We performed lipidomics on isolated dermal adipocytes from Ctrl- or Afa-treated rats to characterize the composition of fatty acids (FAs). The fatty acid composition of adipocytes from Afa- or Ctrl-treated rats did not show any significant differences, and the only significantly changed medium-chain FA was decanoic acid (10:0) (*Figure 4—figure supplement 1B, C*). Notably, the most abundant FAs were mainly C18 fatty acids, including methyl oleate (18:1), methyl elaidate (18:1T), methyl linoleate (18:2) and methyl linoelaidate (18:2T) (*Figure 4—figure supplement 1C*), indicating the extensive release of these types of FAs to the cell matrix during lipolysis. Correspondingly, the free fatty acid (FFA) in the skin homogenates showed an enrichment in Afa-treated rats (*Figure 4—figure supplement 1D*). Experimental studies support the hypothesis that lipolysis-induced lipid accumulation drives macrophage infiltration (*Kosteli et al., 2010*; *Shi et al., 2006*). Lipolysis and a decrease in adipocyte size corresponded with the infiltration of macrophages when rash occurred (*Figure 4D and E*). We hypothesized that individual FA components might promote the migration of macrophages. Monocytes have been well known migrated to local inflammatory sites and differentiated to macrophages. Thus, we treated human monocyte THP-1 cells with one significantly changed FA, decanoic acid (10:0), or the other four most abundant C18 FAs to verify this hypothesis. In particular, we observed that the number of migrated monocytes was increased in the 18:2-treated wells (*Figure 4—figure supplement 1E, F*).

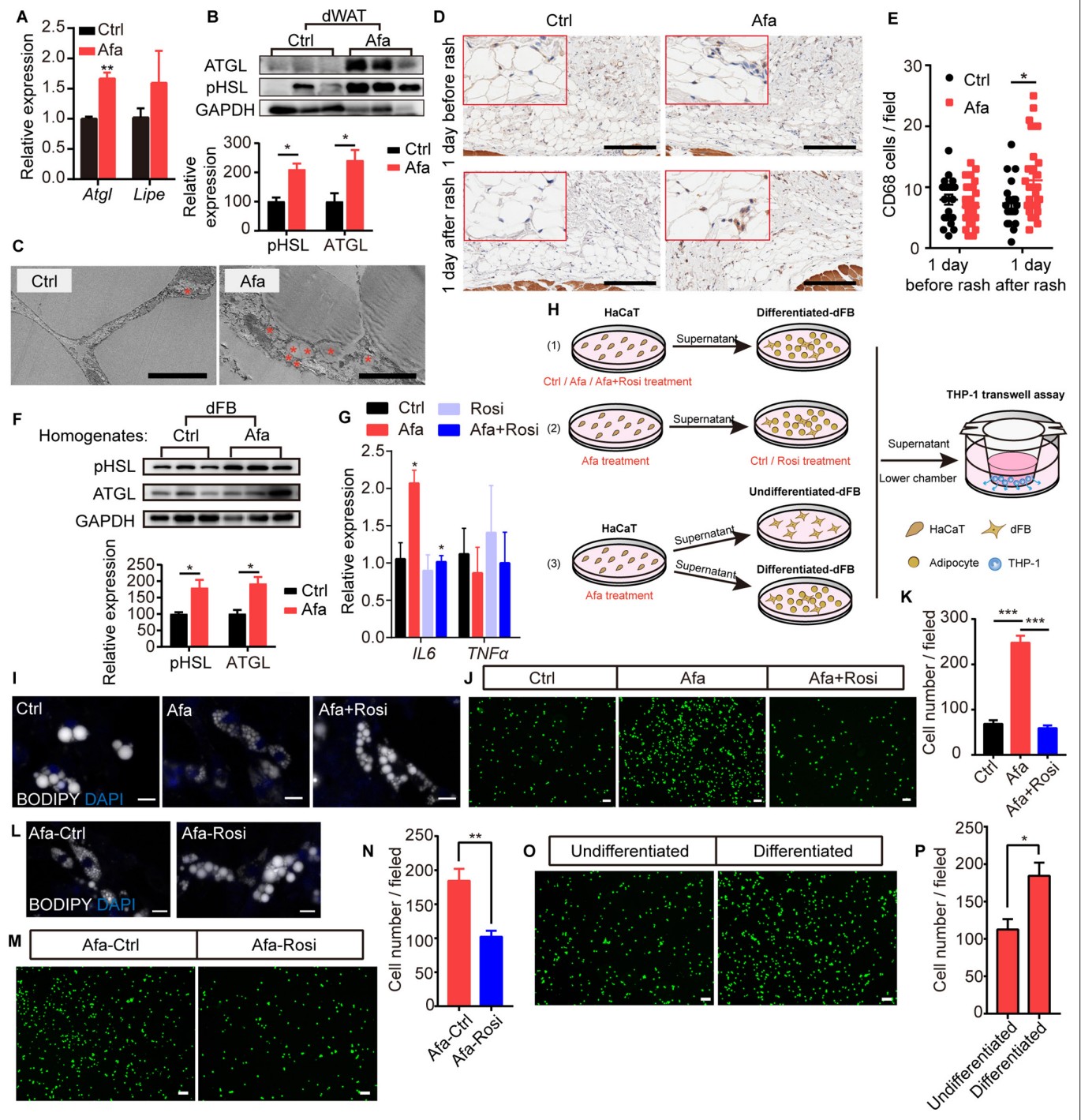

**Figure 4.** Rosiglitazone alleviates lipolysis and inflammatory response. (**A**) Relative expression of the *Atgl* and *Lipe (Hsl)* mRNAs in isolated dWAT on day 5. n = 3 per group. (**B**) Western blot images and quantifications of lipase level from isolated dWAT cells at Grade 1. n = 3. (**C**) Transmission electron microscopy images of dermal adipocytes from Ctrl- and Afa-treated rat skin on day 5. Asterisks show small lipid droplets. Scale bars: 5 µm. (**D and E**) Representative images of CD68 immunostaining (**D**) and quantification (**E**) in skin from Ctrl- and Afa-treated rats at indicated times. Scale bars: 200 µm. (**F**) Western blot images and quantifications showing lipase levels in dFB-derived adipocytes stimulated by rat skin homogenates. n = 3 per group. (**G**) Relative expression of the *IL6* and *TNFα* mRNAs in HaCaT cells after 4 hr of treatment with 100 nM Afa. n = 3 per group. (**H**) Schematic diagram of the experimental setup used to evaluate the roles of Rosi and adipocytes. HaCaT cells were treated with drugs for 24 h, then the supernatants were transferred directly to dFBs for another 24 hr incubation. For transwell assay, the supernatants of dFBs were added to the low chamber. (**I**) Confocal microscopic images of differentiated-dFBs after treatments of HaCaT supernatants from Ctrl, Afa and Afa +Rosi. Lipids were stained with BODIPY 493/503. Scale bars: 50 µm. (**J and K**) Representative images (**J**) and quantification (**K**) of migrated THP-1 cells stimulated with

*Figure 4 continued on next page*

*Figure 4 continued*

culture media from HaCaT supernatants (Ctrl, Afa, Afa +Rosi)-treated dFB cells. (**L**) Confocal microscopic images of HaCaT supernatant (Afa)-treated dFBs after treatment with Ctrl or Rosi. Scale bars: 50 µm. (**M and N**) Representative images (**M**) and quantification (**N**) of migrated THP-1 cells stimulated with culture media from HaCaT supernatant (Afa)-treated dFB cells. Then, 5 µM Rosi or vehicle was added to dFBs. (**O and P**) Representative images (**O**) and quantification (**P**) of migrated THP-1 cells stimulated with culture media from HaCaT supernatant (Afa)-treated undifferentiated or differentiated dFB cells. Data are presented as the means ± SEM. *p < 0.05, **p < 0.01, and ***p < 0.001 (Afa/Rosi group *vs* Ctrl group and Afa +Rosi group *vs* Afa group) using two-tailed unpaired Student's t test and one-way ANOVA.

The online version of this article includes the following source data and figure supplement(s) for figure 4:

**Source data 1.** Original western blot data for *Figure 4B*.

**Source data 2.** Original western blot data for *Figure 4F*.

**Figure supplement 1.** Lipolytic effects of direct EGFRI stimulation on adipocytes.

**Figure supplement 1—source data 1.** Original western blot data for *Figure 4—figure supplement 1G*.

**Figure supplement 1—source data 2.** Original western blot data for *Figure 4—figure supplement 1M*.

---

To further study how EGFR inhibition promotes adipocyte lipolysis, we performed a direct EGFRI treatment on dFB-derived adipocytes. Unexpectedly, direct treatment of adipocytes with Afa did not increase lipase expression or lipolysis-induced glycerol production (*Figure 4—figure supplement 1G, H*). In addition, we detected a multitude of small lipid droplets that were newly formed in the isoproterenol group as a positive control, whereas no differences were detected between the Afa-treated and BSA-treated basal groups (*Figure 4—figure supplement 1I*). Many studies have reported that the high levels of inflammatory cytokines released during inflammatory diseases increase lipolysis to support the immune response (*Feingold et al., 1992*). In individuals with EGFRI-induced skin toxicities, the levels of various inflammatory cytokines are increased, including IL6, TNFα, and CCL2 (*Lichtenberger et al., 2013*; *Mascia et al., 2013*). TNFα and IL6 are known to increase lipolysis in adipocytes (*Feingold et al., 1992*; *Pedersen et al., 2003*). In contrast, we did not observe differences in the expression of these genes in dWAT cells isolated from the skin of Ctrl- and Afa-treated rats (*Figure 4—figure supplement 1J*). However, treatment of adipocytes by homogenates from rat skin revealed obvious upregulation of lipase activity (*Figure 4F*), thus indicating that cytokines secreted by other skin cells might contribute to lipolysis. According to recent studies, EGFR-inhibited or EGFR-depleted skin epidermal keratinocytes, regulate key factors involved in skin inflammation (*Lichtenberger et al., 2013*; *Mascia et al., 2013*). We doubted whether epidermal HaCaT cells regulated adipose lipolysis by releasing inflammatory cytokines. Consistent with previous findings, Afa-stimulated HaCaT cells exhibited higher expression levels of *IL6*, but not *TNFα* (*Figure 4G*). Notably, the addition of Rosi exerted an obvious inhibitory effect on the transcription of *IL6* (*Figure 4G*). We designed several coculture experiments to characterize the roles of Afa and Rosi on adipocytes in inflammation and lipolysis (*Figure 4H*). First, we added Afa to epidermal HaCaT cells to stimulate *IL6* transcription, then treated differentiated-dFBs with the supernatants from HaCaT cells, and finally collected dFB supernatants for monocyte chemotactic assays. Multiple small lipid droplets formed in the Afa-treated dFB, while larger lipids maintained in the Ctrl group (*Figure 4I*). The number of migrated monocytes was also significantly increased in the Afa group (*Figure 4J and K*). Notably, addition of Rosi simultaneously with Afa suppressed both the lipolysis and monocyte migration (*Figure 4I–K*), indicating that the production of inflammatory cytokines was inhibited by Rosi at the initial stage. Consistent with the transcriptional level, secreted IL6 was increased after Afa treatment both in rat skin homogenates and HaCaT supernatants, and decreased with the addition of Rosi (*Figure 4—figure supplement 1K, L*). We next performed the stimulation of IL6 on differentiated-dFB, the lipolysis enhanced by increased level of lipase (*Figure 4—figure supplement 1M*), suggesting that the IL6 secreted by keratinocytes induced lipolysis of adipocytes.

Activation of PPARγ was reported to stimulate lipid reabsorption, reduce FFA secretion and improve lipid homeostasis (*Guan et al., 2002*; *Sun et al., 2012*). To evaluate the direct effect of Rosi on adipocytes during lipolysis, we treated adipocytes with Rosi in the presence of supernatant from Afa-treated HaCaT cells. The lipid stability was improved with the existence of Rosi (*Figure 4L*). Although not as significant as the results shown in *Figure 4J and K*, the elevated chemotaxis of monocytes was also attenuated by Rosi (*Figure 4M and N*). Given the importance of lipolysis in inflammation, we then dissected whether the participation of adipocytes amplified the inflammatory

response. Compared with stimulation of undifferentiated dFBs, the induction of inflammation was remarkable in differentiated dFBs (*Figure 4O and P*). Collectively, these data suggest that the EGFRI induced dermal adipocyte lipolysis, whereas the production of lipolytic cytokines was not induced by the direct stimulation of adipocytes but occurred in keratinocytes. The increased levels of FAs, especially 18:2, are likely to induce monocyte migration. Rosi treatment alleviated lipolysis and the inflammatory response by acting on both the epidermis and adipocytes.

## Expansion and ablation of dWAT reveal its important roles in the rash process

To directly examine the roles of dermal adipocytes in skin rash during EGFR inhibition while avoiding systemic obese phenotypes or loss of other adipocytes, we established a short-term HFD-induced dWAT expansion model and a pharmacological dWAT ablation model.

Recent reports described the substantial expansion of the dWAT layer in HFD-fed obese mice *Kasza et al., 2016*; additionally, increased proliferation of total stromal vascular fraction cells in WAT after long-term HFD feeding and activation of APs at the onset of obesity have been reported (*Jeffery et al., 2015*). To weaken the metabolic disorders in the obese state, we administered a short-term HFD diet and subsequent EGFRI treatment. Rats fed an HFD for 5 weeks showed an expansion of the dermal adipose layer and a significant increase in the adipocyte size (*Figure 5A–C*), while no differences were observed in other adipose depots (*Figure 5D*), body weight changes (*Figure 5E*) or systemic serum lipid levels (*Figure 5—figure supplement 1A*). We analyzed the expression of general adipogenic genes and AKT signaling in the dFB of rats after 5 weeks of HFD feeding to examine whether AP activation was increased. Significant increases in adipogenesis and AKT phosphorylation were observed (*Figure 5F and G*), verifying an increase in adipogenic ability stimulated by the HFD. Notably, the HFD group showed a reduced occurrence and exacerbation of rash (*Figure 5H and I*; *Figure 5—figure supplement 1B*). This result is consistent with a thicker dermal adipose layer and abundant lipid staining (*Figure 5J*). In addition, rats fed an HFD displayed lower white blood cell, granulocyte and lymphocyte percentages (*Supplementary file 2*), the skin infiltration of various immune cells, including mast cell, macrophage, T cell, and monocyte were also decreased (*Figure 5— figure supplement 1C*). The expression of inflammatory receptors and their ligands were also lower than that in Ctrl-treated rats, suggesting a reduced inflammatory response (*Figure 5—figure supplement 1D*, E).

Deoxycholic acid (DCA) injection is a cytolytic agent approved by the FDA to reduce subcutaneous fat (*FDA, 1988*). The active ingredient DCA is structurally identical to endogenous deoxycholate, which serves to solubilize dietary diet; therefore, DCA is designed to disrupt the lipid bilayer of cell membranes and induce cell death (*Schuller-Petrovic et al., 2008*). Based on the function of DCA, we established a dWAT ablation model by administering an intradermal DCA injection. Using female SD rats, we analyzed the effect of DCA after nine total repeated intradermal injections (*Figure 5— figure supplement 2A*). Although the thickness of dWAT did not decrease (*Figure 5—figure supplement 2B, C*), the size of adipocytes was obviously reduced (*Figure 5—figure supplement 2B, D*). In addition, other skin-related adipocyte depots, such as sWAT, iWAT, and BAT, remained unchanged (*Figure 5—figure supplement 2E*). After dWAT ablation, we examined the skin rash process induced by Afa oral gavage in control and DCA rats. The ablation of dWAT didn't affect body weight or food intake (*Figure 5—figure supplement 2F, G*), but resulted in a moderate deterioration in rash severity after 8 days (*Figure 5—figure supplement 2H-J*). A routine analysis of white blood cells also showed an elevation in the DCA group (*Supplementary file 3*). Taken together, we conclude that dermal adipocytes exert potent protective effects on maintaining skin homoeostasis during EGFR inhibition, and the expansion of dWAT is an effective intervention to attenuate skin toxicities. Considering the superficial location of dWAT, a reasonable speculation is that topical application of Rosi would induce the expansion of dWAT, promote AP differentiation and suppress inflammatory cytokine transcription, therefore relieving the skin toxicities of EGFRIs.

## Rosiglitazone application improves the skin phenotype

We topically applied a Rosi gel and its vehicle gel on shaved backs of rats daily after Afa gavage to assess the prophylactic effect of Rosi on skin toxicity during Afa treatment. Modulating dWAT by the topical application of Rosi gel caused obvious improvements in skin phenotypes (*Figure 6A*),

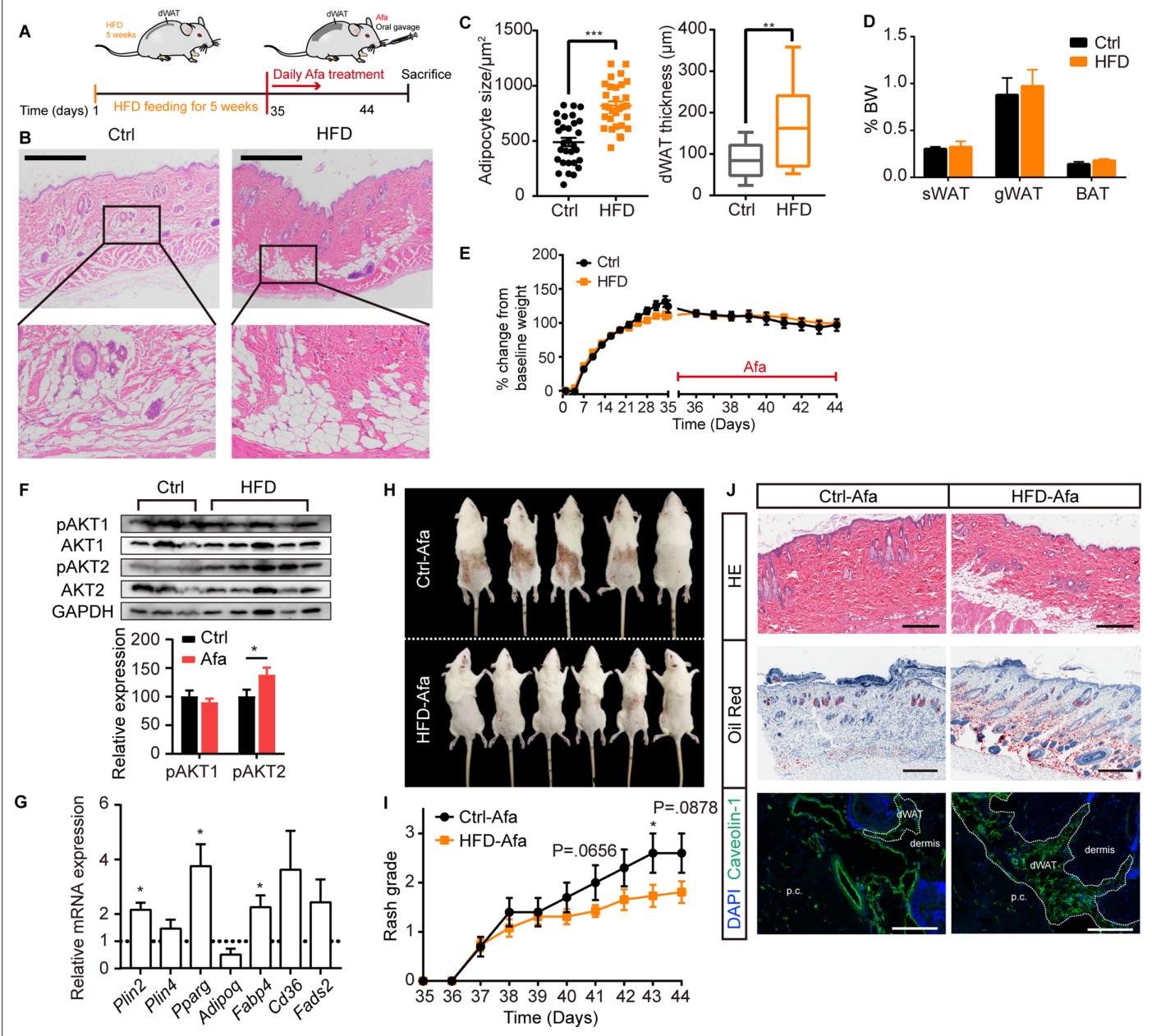

**Figure 5.** HFD-induced dWAT expansion ameliorates rash phenotypes. (**A**) Schematic of the strategy used to expand dWAT by the short-term administration of a HFD. (**B**) H&E staining of skin from normal diet- and HFD-fed rats. Scale bars: 600 μm. (**C**) Size and thickness of dermal adipocytes. (**D**) Percentage of sWAT, gWAT and BAT relative to BW. (**E**) Body weight change. (**F**) Western blot analysis and quantifications of isolated-dWAT cells. n = 3–5 per group. (**G**) Relative mRNA expression of adipogenic genes of isolated-dWAT cells. n = 5 per group. (**H**) Photos of the rash at 44 days. (**I**) Changes in the rash grade after Afa treatment. (**J**) H&E, Oil Red and Caveolin-1 staining of skin biopsies from Ctrl and HFD rats. Scale bars: 300, 500 and 130 μm (top to bottom). Data are presented as the means ± SEM. *p < 0.05, **p < 0.01, and ***p < 0.001 (HFD group *vs* Ctrl group) using two-tailed unpaired Student's t test.

The online version of this article includes the following source data and figure supplement(s) for figure 5:

**Source data 1.** Original western blot data for *Figure 5F*.

**Figure supplement 1.** Additional characterization of HFD rats.

**Figure supplement 2.** DCA-induced dWAT ablation aggravates rash phenotypes.

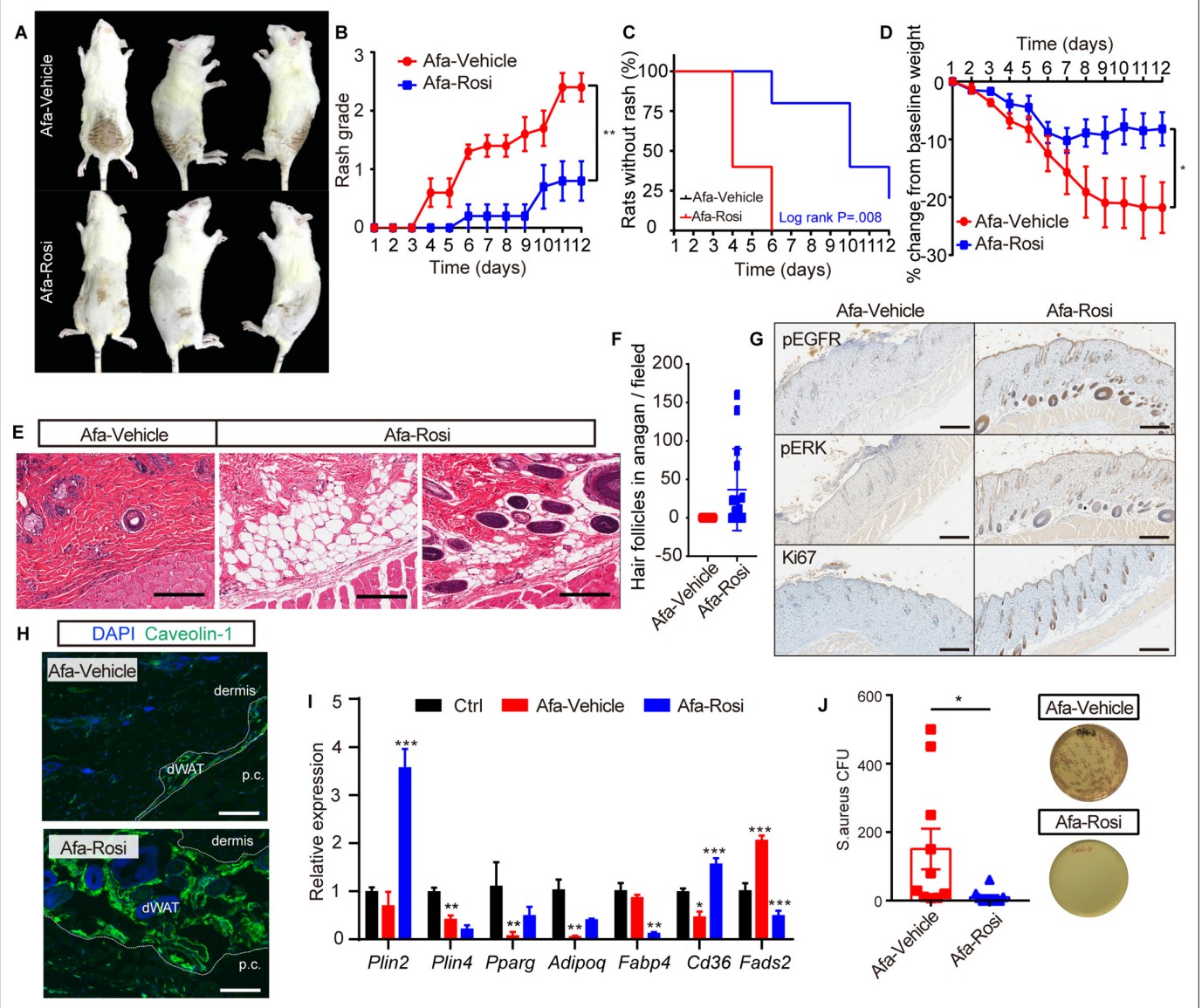

**Figure 6.** Prophylactic application of Rosi prevents skin toxicities. (**A**) Representative photos of rash from Vehicle- and Rosi-treated rats. (**B**) Rash grade. (**C**) Rash occurrence. (**D**) Body weight change. (**E**) H&E staining of skin biopsies. Scale bars: 200 μm. (**F**) Number of hair follicles in anagen phase. (**G**) Immunostaining for pEGFR, pERK, and Ki67 in skin biopsies from Vehicle- and Rosi-treated rats. n = 3 per group. Scale bars: 500 μm. (**H**) Immunostaining for Caveolin-1. Scale bars: 130 μm. (**I**) Relative expression of adipogenic genes in isolated dWAT cells after Vehicle or Rosi treatment. n = 3. (**J**) *S. aureus* growth in media supplemented with skin homogenates from Vehicle- or Rosi-treated rats. n = 5 per group. Data are presented as the means ± SEM. *$p < 0.05$, **$p < 0.01$, and ***$p < 0.001$ (Rosi group *vs* Vehicle group) using two-tailed unpaired Student's t test.

The online version of this article includes the following source data and figure supplement(s) for figure 6:

**Figure supplement 1.** Additional effects of Rosi prevention in Afa-treated rats.

**Figure supplement 1—source data 1.** Original western blot data for **Figure 6—figure supplement 1B**.

**Figure supplement 2.** Therapeutic application of Rosi ameliorates skin rash.

**Figure supplement 3.** Rosi did not interfere the anti-tumor effect of EGFRI.

with a delayed process and low occurrence of rash (**Figure 6B and C**), and improvements in body weight changes (**Figure 6D**). The architecture of Rosi-treated skin showed fewer neutrophil microcysts, less vascular dilatation and congestion, and more parallel growing hair follicles in anagen phase (**Figure 6E–G**). Specifically, the dermal adipocyte layer was still retained in the Rosi group (**Figure 6E and H**). The adipogenic ability recovered (**Figure 6I**; **Figure 6—figure supplement 1A**), and lipolysis decreased after Rosi treatment (**Figure 6—figure supplement 1B**). Consistent with our previous

investigations, the number of APs increased after Rosi treatment (*Figure 6—figure supplement 1C, D*), accompanied by an increased ability to defend against *S. aureus* (*Figure 6J*). Additionally, the application of Rosi also improved serum lipid levels (*Figure 6—figure supplement 1E*). The increased expression level of inflammatory cytokines was downregulated (*Figure 6—figure supplement 1F*), and the immune infiltration was attenuated obviously in Rosi-treated group (*Figure 6—figure supplement 1G, H*). To further verify Rosi's efficacy under therapeutic treatment setting, we applied Vehicle or Rosi gel after rats developed Grade one rash, the rash severity and body weight lose were both significantly alleviated by therapeutic treatment of Rosi (*Figure 6—figure supplement 2*) compared with vehicle treatment.

Encouraged by the prophylactic and therapeutic efficacy of Rosi in ameliorating Afa-induced skin toxicity, we next evaluated whether Rosi interferes with the antitumor efficacy of the EGFRI. We topically applied Rosi gel to nude mice subcutaneously receiving PC9 lung carcinoma cells, for which EGFR inhibitors are approved for first-line treatment. After the gavage of Afa, tumor volumes decreased significantly in both the vehicle gel and Rosi gel groups, and the tumor inhibition rates showed no change between vehicle and Rosi groups; in contrast, control mice treated with vehicle displayed rapid tumor growth (*Figure 6—figure supplement 3A-D*). Intriguingly, Rosi improved body weight and tolerance to Afa (*Figure 6—figure supplement 3E*), and mortality occurred in the Ctrl and Afa groups, with six mice in each group. Taken together, these results clearly show that Rosi regulates dWAT to ameliorate EGFRI-associated cutaneous toxicity without interfering with the antitumor effect.

## Discussion

Cutaneous toxicities induced by anti-EGFR therapy, in particular rash, are the most frequent adverse effects observed in patients with EGFR-mutant cancers, including non-small cell lung cancer, breast cancer and colorectal cancer. The severity of rash is commonly considered a factor associated with a better prognosis (*Peréz-Soler and Saltz, 2005*). The pathogenesis of rash is multifactorial. Recent clinical and experimental findings have described EGFRI-induced rash as an inflammatory disease. Several epidermis-targeted genetic mouse models have revealed the relationship of epidermal EGFR with local immune cell activation, keratinocyte differentiation, hair eruption and microbiota outgrowth (*Klufa et al., 2019*; *Lichtenberger et al., 2013*; *Mascia et al., 2013*; *Satoh et al., 2020*). Mice lacking EGFR in the epidermis developed the same phenotypes of skin lesions as patients treated with EGFRIs: neutrophilic pustules, initial infiltration of macrophages and mast cells, subsequent T cell infiltration, hair follicle destruction, keratin plugs and bacterial invasion (*Mascia et al., 2013*). In humans and mice, the recruitment of inflammatory cells is attributed to the production of proinflammatory chemokines such as CCL5 and CCL2 (*Lichtenberger et al., 2013*; *Mascia et al., 2013*). However, crossing EGFR$^{\Delta ep}$ mice with CCR2$^{-/-}$, MyD88$^{-/-}$, TNFR$^{-/-}$ or NOS2$^{-/-}$ mice did not ameliorate the skin phenotype, while the depletion of macrophages or mast cells only slightly improved the skin phenotype (*Lichtenberger et al., 2013*; *Mascia et al., 2013*). These results reveal the complexity of the inflammatory response and various cell-cell interactions following EGFR blockade. The interference of a single-cell type or a particular signaling pathway may not be sufficient to explain the abnormal skin phenotypes, which include hair follicles, sebaceous glands, dermal immune cells, and blood vessels. Current reports described dermal adipose tissue as a composite tissue maintaining skin functions. dWAT participates in hair cycling, wound healing, thermogenesis, fibrosis and scarring, and immune modulation. Notably, we first noticed that the dWAT layer was diminished in animal models with either genetic EGFR ablation or pharmacological inhibition, and hair follicles were arrested in catagen phase and located far away from the dWAT layer. However, researchers have not determined how dermal adipocytes respond to EGFR inhibition. Moreover, the effect of modulating dWAT on this pathological condition requires further investigation.

Here, we show that a reduction in dermal adipocytes is an early response to EGFR inhibition and leads to the disorder of cutaneous homeostasis (*Figure 7*). Impaired adipogenesis decreased the skin defense against *S. aureus*, as well as the signaling interactions with hair follicles. In addition, increased migration of monocytes was associated with lipolysis in adipocytes characterized by increased lipase activity. Activation of lipolysis in dermal fat occurs as a response to skin wound repair (*Schmidt and Horsley, 2013*; *Shook et al., 2020*). Although macrophages recruited during wound healing by lipolysis were reported to promote revascularization and re-epithelialization (*Shook et al.,*

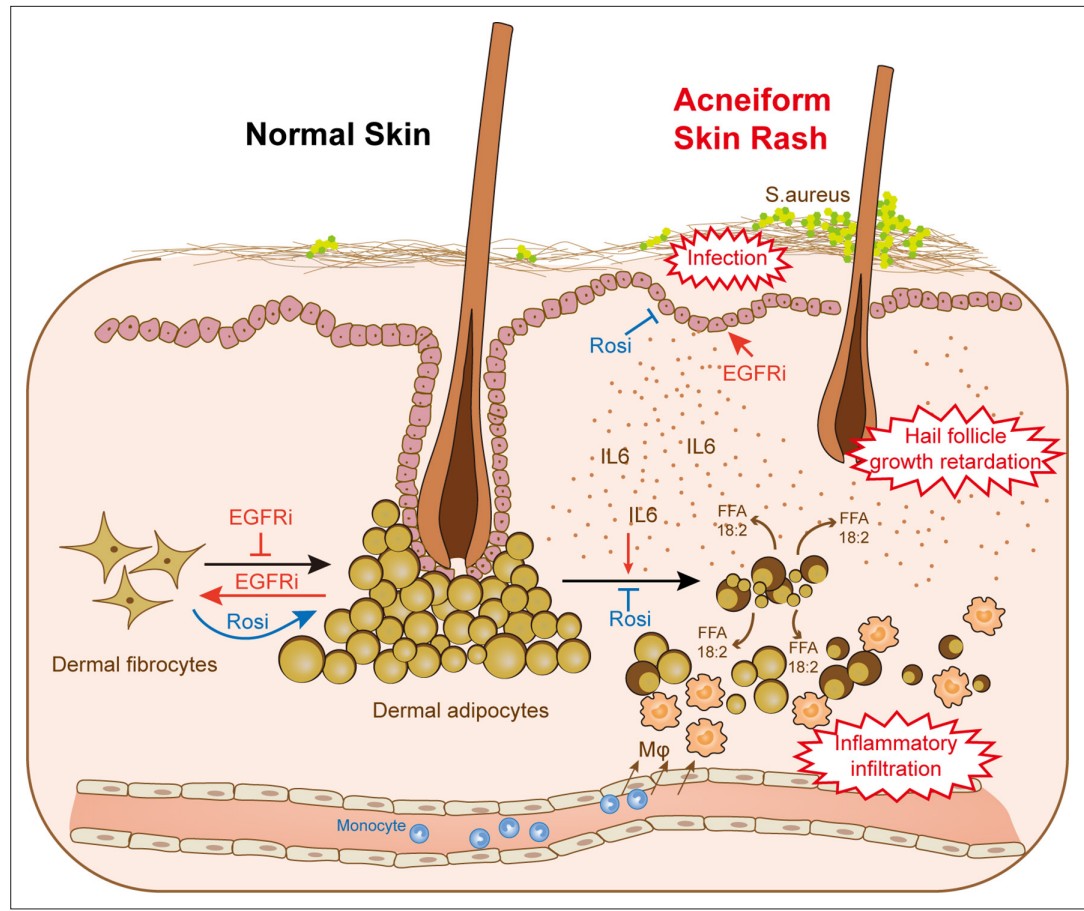

**Figure 7.** dWAT contributes to skin homeostasis during anti-EGFR therapy. Reduction of dWAT is a hallmark of EGFRi-induced cutaneous toxicity. EGFR inhibition blocks adipocyte progenitor differentiation and induces dedifferentiation of mature dermal adipocytes. In addition, lipolysis is activated by the up-regulation of the lipolytic cytokine IL6 through epidermal keratinocytes, leading to the infiltration of monocyte-derived macrophages. Lack of dWAT also impairs skin host defense and hair follicle growth. Activation of PPARγ by Rosi could both alleviate the block of differentiation and suppress the increased lipolysis through transcription inhibition of IL6, thus relieving the abnormal skin phenotypes in vivo.

*2020*), increased macrophage numbers and free fatty acid contents may also stimulate inflammatory responses via various pattern recognition receptors and cytokine receptors in adipocytes, subsequently inducing cytokine expression and increasing chemoattractant activity (*Shi et al., 2006*; *Wang et al., 2009*; *Zeng et al., 2018*). The infiltration of macrophages in adipose tissue coincided with circulating FFA concentrations and adipose lipolysis, and pharmacologically induced lipolysis by a β3-adrenergic agonist increased macrophage accumulation (*Kosteli et al., 2010*). Correspondingly, our data also showed a causal relationship among macrophage infiltration, FFA secretion and lipolysis following EGFRI treatment (*Figure 4C and D*; *Figure 4—figure supplement 1B, C*), and the lipids released by disrupting adipocyte cell membranes in the DCA model indicated an accumulation of macrophages (*Figure 5—figure supplement 2J*). Notably, we showed that dermal adipocytes amplified inflammatory responses from epithelial cells primed with EGFRI treatment. Although epithelial keratinocytes alone secreted chemokines and induced inflammatory infiltration, the participation of dermal fat enhanced this process (*Figure 4M and N*). The ability of dermal lipids to costimulate keratinocyte-generated cellular responses represents a novel pathway for the amplification of innate immunity and inflammation. Based on these data, local lipid fluxes are central regulators of the inflammatory response in EGFRI-affected dermal adipose tissue. In this study, we also found a distinction between dermal and subcutaneous white adipose tissue. Although both of these tissues respond early and decrease in size after EGFR inhibition, the expression of adipogenic and fibrotic genes shows different trends; additionally, lipolysis does not occur in subcutaneous fat. These results

indicate that the dermal adipose layer is distinct from subcutaneous adipocytes and has a unique role in skin modulation.

Innate immunity of dermal adipose tissue mediated by the release of antimicrobial peptides protects many organs from bacterial infection, such as the skin, peritoneum, endocardium and colon (*Dokoshi et al., 2018*; *Schmid et al., 2017*; *Zhang et al., 2015*). Impaired adipogenesis induced by the PPARγ inhibitors GW9662 and BADGE results in an increased bacterial infection in the skin and colon (*Dokoshi et al., 2018*; *Zhang et al., 2015*). Moreover, a recent study observed that retinoids enhance and sustain Camp levels in developing preadipocytes (*Liggins et al., 2019*). Interestingly, retinoids are therapeutically useful against acne, a disorder driven partly by bacteria (*Chivot, 2005*). This correlation might also be beneficial to treat acneiform lesions in patients with EGFRI-associated skin disorders. Another skin-related disorder called scarring alopecia is caused by a specific *Pparg* depletion in hair follicle stem cells, and immunohistochemical staining showed proinflammatory lipid and cell accumulation (*Karnik et al., 2009*). The interplay between dWAT and hair follicles also involves the modulation of the stem cell niche. Follicular stem cell activation was verified to be associated with dermal adipocytes, and hair follicles of mice treated with BADGE or GW9662 did not enter anagen phase and were blocked in telogen phase (*Rivera-Gonzalez et al., 2016*).

PPARγ acts directly to negatively regulate the expression of inflammatory genes in a ligand-dependent manner by antagonizing the activities of transcription factors, such as members of the NF-κB and AP-1 families. PPARγ stimulation in rodents has been shown to ameliorate several inflammatory diseases, such as atopic dermatitis, psoriasis and acne vulgaris (*Ramot et al., 2015*). However, previous studies did not obtain evidence of the effect of PPARγ modulation on epidermal functions perturbed by an EGFRI. Here, we show that dWAT plays a critical role in the pathogenesis of cutaneous toxicities induced by EGFR inhibition. Experiments with HFD feeding and DCA injection have shown that dWAT is required to maintain skin homeostasis during EGFR inhibition. Indeed, topical application of rosiglitazone to EGFRI-treated rats prevents the onset of rash and ameliorates the symptoms. Additionally, based on the results obtained from the tumor-bearing nude mice, topical application of rosiglitazone gel did not alter the antitumor effect of afatinib. Therefore, PPARγ agonists may represent a promising new therapeutic strategy in the treatment of EGFRI-related skin disorders.

# Materials and methods

## Key resources table

| Reagent type (species) or resource | Designation | Source or reference | Identifiers | Additional information |
|---|---|---|---|---|
| Cell line (*Homo sapiens*) | HaCaT | Cell Research | Cat. #: ZQ0044 | |
| Cell line (*Homo sapiens*) | PC-9 | Cell Bank of Chinese Academy of Sciences | Cat. #: SCSP-5085 CSTR:19375.09.3101HUMSCSP5085 | |
| Cell line (*Homo sapiens*) | THP-1 | Cell Bank of Chinese Academy of Sciences | Cat. #: TCHu 57 CSTR:19375.09.3101HUMTCHu57 | |
| Antibody | Rabbit monoclonal anti-ATGL | Abcam | Cat. #: ab109251 | WB (1:1000) |
| Antibody | Rabbit monoclonal anti-caveolin-1 | CST | Cat. #: 3,267 | WB (1:1000) |
| Antibody | Rabbit monoclonal anti-phoso-AKT1 | CST | Cat. #: 5,012 | WB (1:1000) |
| Antibody | Rabbit monoclonal anti-phoso-AKT2 | CST | Cat. #: 8,599 | WB (1:1000) |
| Antibody | Rabbit polyclonal anti- phospho-HSL | Absin | Cat. #: abs139855 | WB (1:1000) |
| Antibody | PE Mouse Monoclonal Anti-rat CD31 | BD Pharmingen | Cat. #: 555,027 RRID: AB_395657 | FACS (0.25 µg per million cells in 100 µl volume) |
| Antibody | FITC Mouse Monoclonal anti-rat CD45 | BioLegend | Cat. #: 202,205 RRIS: AB_314005 | FACS (0.25 µg per million cells in 100 µl volume) |
| Antibody | PerCP/Cyanine5.5 Armenian Hamster Monoclonal Anti-mouse/rat CD29 | BioLegend | Cat. #: 102,227 RRIS: AB_2572078 | FACS (0.25 µg per million cells in 100 µl volume) |

*Continued on next page*

*Continued*

| Reagent type (species) or resource | Designation | Source or reference | Identifiers | Additional information |
|---|---|---|---|---|
| Antibody | Pe-Cy7 Mouse Monoclonal Anti-mouse/rat/human CD34 | Santa Cruz Biotechnology | Cat. #: sc-7324 RRIS: AB_2572078 | FACS (1 µg per million cells in 100 µl volume) |
| Peptide, recombinant protein | Recombinant human insulin | Sigma | Cat. #: 407,709 | |
| Commercial assay or kit | NEFA LabAssay | Wako Diagnostics | Cat. #: 294–63601 | |
| Chemical compound, drug | Liberase TL | Roche | Cat. #: 05401020001 | |
| Chemical compound, drug | Dexamethasone | Sigma | Cat. #: D4902 | |
| Chemical compound, drug | Indomethacin | Sigma | Cat. #: I8280 | |
| Chemical compound, drug | 3-isobutyl-1-methylxanthine (IBMX) | Sigma | Cat. #: I7018 | |
| Chemical compound, drug | BODIPY 493/503 | ThermoFisher | Cat. #: D3922 | |
| Software, algorithm | Aperio ImageScope | Leica | | |

## Animal models and treatment

Female 6–8 weeks SD rats and neonatal rats were obtained from Jihui, Shanghai, China. Female 3 week SD rats were purchased from Charles River, Beijing, China. Six week BALB/c nude mice were from Jihui, Shanghai, China. Rash model was generated by daily p.o. of Afatinib at a dose of 40 mg/kg for about 10 consecutive days, the back hair was gently shaved for observation. According to our statistical data (*Figure 1—figure supplement 1C*), the mean time for rash occurrence is day 4, so we sacrificed rats one day before rash at day 3, and 1 day after rash at day 5.

For HFD experiment, 3 week rats were fed with an HFD diet for 5 weeks to induce the expansion of the dWAT layer, then followed with Afatinib treatment to construct rash model. For rosiglitazone application experiment, vehicle gel or rosiglitazone gel (made of propylene glycol, transcutol, PEG400, water, carbopol 980% and 1% rosiglitazone) was applied topically on the shaved back of rat, the rats were fixed in a cylindrical holder for four hours during the application, and then the residual gel was removed. The back skin changes were visually inspected every day.

For dWAT ablation experiment, deoxycholic acid (vehicle: 0.9% benzyl alcohol in PBS) was intradermal injected in rat back skin every 3 days, each rat received 2.5 mg/cm$^2$ deoxycholic acid for nine times.

For prophylactic and therapeutic Rosi treatment, Afatinib was administered p.o. once daily at dose level of 40 mg/kg, Vehicle or Rosi gel was applied topically when the Afatinib treatment started or after Grade one rash occurred. The rats were fixed in a cylindrical holder for 4 hr during the application of the gel, and then the residual gel was removed. The changes of back skin were visually inspected every day.

To develop xenograft tumor transplanting mice model, a single-cell suspension of $5 \times 10^6$ PC9 cells was inoculated subcutaneously into each nude mouse. After 3 weeks, oral gavage of 30 mg/kg Afatinib and topical application of vehicle or rosiglitazone gel were started when tumor volume reached 100 mm$^3$.

## Reagents

Afatinib, erlotinib, gefitinib, osimertinib, and rosiglitazone were purchased from Goyic, China. Propylene glycol was from ER-KANG, China. Transcutol was from GATTEFOSSé, French. Carbopol 980 was from 2Y-Chem, China. PEG400 was from YIPUSHENG, China. Dexamethasone (D4902), indomethacin (I8280), recombinant human insulin (407709) and 3-isobutyl-1-methylxanthine (IBMX) (I7018), Isoproterenol (1351005) were from Sigma. Deoxycholic acid (D2510) was obtained from Sigma. The anti-PDGFR alpha (ab203491) and anti-ATGL (ab109251) antibodies were from Abcam. The anti-caveolin-1 (3267), anti-phospho-AKT1 (5012), anti-AKT1 (2938), anti-phospho-AKT2 (8599), anti-AKT2

(3063) antibodies were from Cell Signaling Technology. The anti-phospho-HSL (abs139855) was from Absin. The anti-HSL antibody (NB110-37253) was from Novus. The GAPDH antibody (60004–1-Ig) was from Proteintech. The anti-IL6 (GB11117), anti-TNFα (GB11188), anti-CD3 (GB11014), anti-CD11b (GB11058), anti-CD68 (GB113109) antibodies were from Servicebio. The anti-PPAR Gamma (16643–1-AP), anti-Perilipin-2 (15294–1-AP), anti-Mcp1(66272–1-Ig) antibodies were from Proteintech. The secondary antibodies used were from Beyotime. BODIPY 493/503 (D3922) was purchased from ThermoFisher. Deoxycholic acid (D2510) was obtained from Sigma. Human IL6 (200–06) was from PeproTech.

## Cell culture

HaCaT (Cell Research; Cat. #: ZQ0044) and PC-9 (Cell Bank of Chinese Academy of Sciences; Cat. #: SCSP-5085) cells were cultured in DMEM/high glucose medium, THP-1(Cell Bank of Chinese Academy of Sciences; Cat. #: TCHu 57) cells were cultured in 1640 complete medium, all supplemented with 10% fetal bovine serum (FBS) and 1% penicillin-streptomycin in an incubator under a humidified atmosphere of 5% CO2 at 37 °C. All cells were tested for microplasma contamination and authenticated by STR profiling.

## Rash grade evaluation

Grade 0-normal; Grade 1-papules and/or pustules covering <10% surface area of shaved-back skin; Grade 2-papules and/or pustules covering 10–30% surface area of shaved-back skin; Grade 3-papules and/or pustules covering >30% surface area of shaved-back skin.

## Histology, immunohistochemistry (IHC), and immunofluorescence (IF)

Rat skin was embedded by paraffin or OCT and sectioned for staining. H&E staining was performed using paraffin sections. Oil red and Ki67 were directly stained using OCT sections. For IHC and IF, paraffin or OCT sections were fixed and blocked, then stained with CD68 or caveolin-1 primary antibody followed by secondary antibodies. Nuclei were counter-stained with DAPI. All images were scanned with a digital pathology scanning system (NANOZOOMER S360, Leica) or Upright-Reverse Fluorecent Microscopy (Revolve, Echo). The size of adipocyte was measured from HE staining by Aperio ImageScope software. After scanning slides, the pen button allowed users to click and drag a free-form shape to enclose an area to analyze, and the annotation window will show the size of the area. The number of adipocytes was mounted in the same zoom window in Aperio ImageScope.

## Dermal adipocytes and primary dermal fibroblasts isolation and culture

Neonatal (P1) and adult rat back skin was cut into small pieces then digested with Liberase TL (Roche) for 1 hr at 37 °C with constant shaking. Cells mixture was first filtered through 70 µm filter to discard tissue pieces, floating dWAT was collected by centrifuging and followed by washing. Cell suspension was then treated with red blood cell lysis and washed with PBS. dFB suspension was obtained by filtering through 40 µm filter. Isolated dFB was cultured in DMEM supplemented with 10% FBS in a humidified incubator at 5% CO$_2$ and 37 °C. Fresh medium was changed daily and only one passage was used for experiment. To induce adipocyte differentiation, post-confluent dFB was switched to adipocyte differentiation medium containing 2 µM Dexamethasone, 250 µM IBMX, 200 µM Indomethacin and 10 µg/mL recombinant human insulin for 9 days. 10 nM Afa or 5 µM Rosi was added in differentiation medium and changed with fresh differentiation medium at day 3, 6, and 9.

dWAT dedifferentiation was demonstrated using ceiling culture. Briefly, isolated dWAT was transferred to an inverted 25 cm$^2$ cell culture flask, completely filled with DMEM supplemented with 20% fetal bovine serum and Afa or Rosi. Adipocytes were monitored daily for cell attachment. After sufficient attachment and occurrence of dedifferentiated adipocytes, medium was removed, replaced with fresh medium and the flask was reinverted.

For confocal microscopy, dFB cells were cultured, differentiated, and treated with HaCaT supernatants in 12-well cell culture slides. Lipids were stainted with BODIPY 493/503. Cell nucleus were stained with stained with DAPI Fluoromount-G (Yeasen). Microscopy was performed on Leica TCS SP8.

## Cell proliferation assay

Cells were cultured in 96-well plate and treated with different agents. Twenty-four hours after treatments, cells were counted using Cell Counting Kit-8 (Yeasen).

## Flow cytometry and analysis (FACS)

Freshly isolated dFB from rat skin was stained with zombie violet viability dye (Biolegend, 423114) to stain dead cells. Cells were then stained with an antibody cocktails containing PE-CD31 (BD, 555027), FITC-CD45 (BioLegend, 202205), PerCP/Cy5.5-CD29 (BioLegend, 102227), and PE-Cy7-CD34 (Santa-Cruze, sc-7324). FACS analysis for protein expression of each cell marker was performed by the BD FACSCanto RUO machine and analyzed by FlowJo V10 software.

## Transmission electron microscopy (TEM)

TEM was performed in the Instrumental Analysis Center of SJTU. Vehicle and Afa rats were anesthetized after 5 day oral gavaged and immediately transcardially perfused using PBS, then 50 ml of cold 4% PFA. The vehicle control and Afa-induced rash back skin were dissected and cut into small pieces, approximately 1 mm³ in volume. Tissue pieces were fixed in 2.5% glutaraldehyde for 30 min at RT and 1.5 hr at 4 °C, and then rinsed in PBS three times. Samples were postfixed in 1% osmium tetroxide for 1 hr, rinsed and enbloc stained in aqueous 2% uranyl acetate for 1 hr followed by rinsing, dehydrating in an ethanol and acetone series, infiltrating with resin and baking 48 hr at 60 °C. Hardened blocks were cut using an ultramicrotome (Leica EM UC7). Ultrathin 100 nm sections were collected and stained using lead citrate for transmission microscopy. Carbon-coated grids were viewed on a transmission electron microscope (Talos L120C G2) at 120kV.

## Lipolysis assay

For ex vivo lipolysis assay, sWAT explants were collected and cultured in 12-well plate, for in vitro lipolysis assay, differentiated adipocytes were cultured in 24-well plate, cells and explants were cultured in serum-free DMEM containing 2% fatty acid-free BSA. 10 µM Isoproterenol was added to induce lipolysis, cells and explants were treated with Afa. Glycerol was measured after different hours stimulation at 37 °C with shaking. The glycerol assay (Sigma) or NEFA assay (Wako Diagnostics) was used to measure lipolysis, as per manufacturer's instructions.

## Lipid profiling

For quantitative lipid mass spectrometry of adipocyte lipid stores, dWAT of vehicle control and Afa was digested with Liberase TL as described above. Isolated dWAT was washed with HBSS and 10⁶ cells were counted for each sample, total lipids were extracted and FA profiles were quantified by GC/MS.

## *S. aureus* assay

Skin biopsies were homogenized in PBS by Precellys Evolution homogenizer (Bertin; 2 × 30 s at 6000 rpm followed by 30 s on ice after each cycle). A total of 100 µL homogenized skin samples were plated onto *S. aureus* color culture medium (Comagal, France) and counted after 24 hr culture at 37 °C.

## Migration assay

THP-1 migration assay was performed using transwell chamber with 8 µm pore size (Falcon) for 24-well plates. 500 µL of dFB-derived supernatant was loaded in the lower chamber. 200 µL Calcein-AM (Yeasen) –labelled cell suspension was added to the upper chamber. The chamber was incubated at 37 °C for 3 hr in a $CO_2$ incubator. After incubation, THP-1 cells that migrated to the lower chamber were counted under a microscope.

## Western blot

Cells were lysed using RIPA lysis buffer. Proteins were separated on 7.5–10% SDS-PAGE gel and transferred to PVDF membrane. The membranes were blocked for 1 hr in TBST containing 5%BSA, and then incubated with primary antibodies overnight. The membranes were then washed with TBST for 10 min x three times, and incubated in secondary antibodies for 1 hr. Membranes was washed again with TBST for 20 min x three times and revealed using the Super Signal West Pico kit (Thermo Scientific).

## Quantitative PCR

RNA was extracted from rat skin and indicated cells using the TRIzol Reagent (Invitrogen). cDNA was generated from total RNA using a ReverTra Ace qPCR RT Master Mix (FSQ-201, TOYOBO) according

to the manufacturer's instructions. Quantitative real-time PCR was performed with the ABI ViiA7 or ABI 7500 Realtime PCR system using SYBR Green Master Mix (YEASEN). The primers used for amplification of specific genes were synthesized by Sangon (*Supplementary file 4*).

## Enzyme-linked immunosorbent assay

Quantification of Human IL6 protein concentrations in HaCaT culture supernatants, and Rat IL6 from rat skin homogenates were performed by enzyme-linked immunosorbent assay (ELISA) according to the Human/Rat IL6 ELISA kit protocol (Boster).

## Blood biochemistry test

Blood samples were collected into the EDTA-K2 anticoagulative tubes to measure blood routine examination by SYSMEX POCH100IVD. For lipid level examination, blood samples were centrifuged at 3000 g for 10 min to separate serum, the serum lipid were detected by Adicon, Ltd.

## Statistical

Statistical analysis was performed using unpaired Student's $t$ test or one-way analysis ANOVA. Differences were considered significant when p was less than 0.05: $*p < 0.05$; $**p < 0.01$; $***p < 0.001$.

## Acknowledgements

We are grateful to Professor Ling-juan Zhang (School of Pharmaceutical Sciences, Xiamen University, China) for her fruitful discussion. This work was, in part, supported by the Youth Thousand Talents Program of China, start-up grants from the Shanghai Jiao Tong University (WF220408211). This work was also supported by the grants from the State Key Laboratory of Onco- genes and Related Genes (90-17-02) and from the Interdisciplinary Program of Shanghai Jiao Tong University (YG2017MS18). This work was partially supported by the National Natural Science Foundation of China (Grant No. 81773347).

## Additional information

### Competing interests

Shiyi Zhang: is the co-founder of OnQuality Pharmaceuticals. The author has no other competing interests to declare. The other authors declare that no competing interests exist.

### Funding

| Funder | Grant reference number | Author |
|---|---|---|
| National Natural Science Foundation of China | 17Z127060041 、19Z127060001 、19Z127010003 | Shiyi Zhang |
| Shanghai Jiao Tong University | WF220408211 | Shiyi Zhang |
| Shanghai Jiao Tong University | YG2017MS18 | Shiyi Zhang |
| State Key Laboratory of Onco- genes and Related Genes | 90-17-02 | Shiyi Zhang |
| National Natural Science Foundation of China | 81773347 | Nan Xu |

The funders had no role in study design, data collection and interpretation, or the decision to submit the work for publication.

## Author contributions

Leying Chen, Conceptualization, Data curation, Formal analysis, Investigation, Methodology, Project administration, Visualization, Writing - original draft; Qing You, Conceptualization, Data curation, Investigation, Methodology, Project administration; Min Liu, Data curation, Formal analysis, Investigation, Methodology, Project administration; Shuaihu Li, Data curation, Formal analysis, Investigation, Project administration; Zhaoyu Wu, Data curation, Formal analysis; Jiajun Hu, Data curation, Formal analysis, Project administration; Yurui Ma, Data curation, Formal analysis, Investigation; Liangyong Xia, Formal analysis, Software; Ying Zhou, Data curation, Investigation; Nan Xu, Funding acquisition, Investigation, Project administration, Supervision, Validation, Writing – review and editing; Shiyi Zhang, Conceptualization, Funding acquisition, Investigation, Project administration, Resources, Supervision, Validation, Writing – review and editing

## Author ORCIDs

Leying Chen (iD) http://orcid.org/0000-0002-6448-4177
Shuaihu Li (iD) http://orcid.org/0000-0002-2049-7304
Shiyi Zhang (iD) http://orcid.org/0000-0002-9375-5992

## Ethics

The Institutional Animal Care and Use Committee of the Shanghai Jiao Tong University (A2017039) and East China Normal University (R20190601) approved all animal experiments.

## Decision letter and Author response

Decision letter https://doi.org/10.7554/eLife.72443.sa1
Author response https://doi.org/10.7554/eLife.72443.sa2

# Additional files

## Supplementary files

• Supplementary file 1. Weight loss in EGFRI-related clinical trials.

• Supplementary file 2. Blood cell analysis of Vehicle and HFD rats. WBC: white blood cells. W-SCR: WBC-small cell ratio. W-MCR: WBC-middle cell ratio. W-LCR: WBC-large cell ratio. W-SCC: WBC-small cell count. WBC-MCC: WBC-middle cell count. WBC-LCC: WBC-large cell count. RBC: red blood cells. HGB: Haemohlobin. HCT: Haematocrit. MCV: Mean corpuscular volume. MCH: Mean corpuscular haemoglobin. MCHC: Mean corpuscular haemoglobin concentration. RDW-SD: RBC-distribution width standard deviation. RDW-CV: RBC-distribution width variation coefficient. PLT: Platelets. PDW: Platelets distribution width. MPV: Mean platelet volume. P-LCR: Platelet large cell ratio. Data are presented as the means ± SEM. $P < 0.05$ using 2-tailed unpaired Student's t test.

• Supplementary file 3. Blood cell analysis of Vehicle and DCA rats. WBC: white blood cells. W-SCR: WBC-small cell ratio. W-MCR: WBC-middle cell ratio. W-LCR: WBC-large cell ratio. W-SCC: WBC-small cell count. WBC-MCC: WBC-middle cell count. WBC-LCC: WBC-large cell count. RBC: red blood cells. HGB: Haemohlobin. HCT: Haematocrit. MCV: Mean corpuscular volume. MCH: Mean corpuscular haemoglobin. MCHC: Mean corpuscular haemoglobin concentration. RDW-SD: RBC-distribution width standard deviation. RDW-CV: RBC-distribution width variation coefficient. PLT: Platelets. PDW: Platelets distribution width. MPV: Mean platelet volume. P-LCR: Platelet large cell ratio. Data are presented as the means ± SEM. $P < 0.05$ using 2-tailed unpaired Student's t test.

• Supplementary file 4. Primer sequence information.

• Transparent reporting form

## Data availability

All data generated or analysed during this study are included in the manuscript and supporting file; Source data are provided with this paper. Data has been uploaded to Dryad.

The following dataset was generated:

| Author(s) | Year | Dataset title | Dataset URL | Database and Identifier |
|---|---|---|---|---|
| Zhang S | 2021 | Data From: Remodeling of dermal adipose tissue alleviates cutaneous toxicity induced by anti-EGFR therapy | https://doi.org/10.5061/dryad.41ns1rndv | Dryad Digital Repository, 10.5061/dryad.41ns1rndv |

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
