## [Editor Report]

This paper will be of interest to oncologists and dermatologists, and has high clinical relevance. It reveals a novel mechanism of EGFR inhibitor-induced rash which be may closely related to atrophy of dermal white adipose tissue (dWAT). A series of experimental manipulations dissect the mechanism with a murine model, supporting the major claims of the paper.

---

## [Decision Letter]

**Decision letter after peer review:**

Thank you for submitting your article "Remodeling of dermal adipose tissue alleviates cutaneous toxicity induced by anti-EGFR therapy" for consideration by *eLife*. Your article has been reviewed by 2 peer reviewers, and the evaluation has been overseen by a Reviewing Editor and Mone Zaidi as the Senior Editor. The reviewers have opted to remain anonymous.

The Reviewing Editor has drafted this to help you prepare a revised submission.

Essential revisions:

Figure 1: A timeline of the development of rash after Afa treatment is missing at the beginning. What is the percentage of rats developing rash grade 1, 2 or 3? Are there any other side effects of Afa treatment (diarrhea, lethality)?

Figure 1: The authors need to better describe their skin inflammation model. Is the rash hair follicle bound? What is the composition and location of the immune cell infiltrate and which cytokines and chemokines are dominant in the skin? Does the rash occur also with an specific EGFR inhibitor instead of the broad ErbB family inhibitor Afa (e.g. Erlotinib). Moreover, does the rash develop only in shaved areas or also in unshaved areas of the back skin? This should be clarified as a recent study associated EGFRi-mediated skin rash development with hair regrowth (Klufa et.)

Figure 1G: The authors show a correlation between rash grade and dWAT thickness. Is there a correlation to body weight? Rats show reduced food intake after Afa treatment (Suppl. Figure 1B). Could this be the cause for the reduction in dWAT? How does dWAT in rats react to fasting?

Does shaving have any effect on Afa induced dWAT reduction? Are similar changes observed in unshaved skin?

Figure 1 E,F, Suppl. Figure 1C,-R: In these figures the labelling is unclear. The authors are labelling the columns Grade 0-3. Have all these animals been harvested on day 10 as mentioned in the methods section? Does grade 0 mean "before" Afa? What do the "control" samples then correspond to, since they also are assigned to rash grades?

Figure 1H-J: PPARγ is strongly downregulated in the skin of Afa treated rats, but the KEGG Pparg signaling pathway is among the top 10 enriched pathways. How do the authors explain this discrepancy? It would be helpful to show all gene changes of the pathway and verify some gene changes on the protein level (e.g.: IHC). Where are these data from? No explanation or citation to a previous study can be found in the manuscript.

Figure 2C and D: How do the authors take samples one day before the rash during the 10 day Afa treatment? How long have the rats been treated with Afa at the 1 day after rash timepoint?

Figure 2E: Authors should show the full timeline of the pictures (also the controls) and provide a quantification of adipocytes from each timepoint.

Figure 3A: These experiments need to be quantified in addition to the representative pictures.

Figure 3C: How often has this experiment been repeated? Some changes are difficult to see. All the Western Blots throughout the manuscript should be quantified to be able to interpret these results: e.g. the claimed reduction of pAkt is not really visible and is also seen in the controls at day 9.

Figure 3D: The population of adipocyte progenitors is not very well demarcated. It has been shown that among the CD34+CD29+ cells in the skin only a subpopulation is adipogenic, and this proportion is also depending on the hair cycle (Festa et al.,). The authors should stain also for CD24 and/or Sca1 to more precisely specify the proportion of adipogenic cells. The same accounts for Figure S7 B, D. (Figure S7D: it does not make sense to depict cells that are not there as MFI = 0):

Figure 3F: The authors claim that *S. aureus* outgrowth is because of impaired adipogenesis (text line 418). This is however an overstatement since a large number of cells (especially keratinocytes) in the skin have been shown to be able to produce antimicrobial factors that might cause this result. S. aureus outgrowth is a hallmark of EGFR-I induced skin inflammation and atopic dermatitis in general. The authors have no mechanistic claim that adipocytes are influencing the bacterial superinfection in this setting. The results are merely a correlation and a comparison of rodents with skin inflammation and without.

Figure 3G: Same as with Figure 3F: This represents a correlation with the general skin inflammation rather than a mechanistic explanation involving the adipocytes. Both statements (Figure 3G and H) should be ideally checked using a adipocyte specific cre line in a mouse model. Furthermore EGFR deletion in keratinocytes alone using K5-cre was enough to induce a dramatic *S. aureus* mediated dysbiosis and hair loss in mice (Klufa et al.,). The loss of adipocytes might still contribute to these effects but are not necessary.

Figure 3H: here EGFR, pEGFR and ERK staining of adipocytes and their respective reduction after Afa treatment should also be shown.

Figure 3I: the quantification is missing. how many HFs have been counted to be able to make this statement?

Figure S4C: In the text (line 444) the authors claim that there is a correlation between high levels of C18 FAs and increased lipolysis without showing any difference in the levels of this FA. This statement should be corrected. The same accounts for the statement in lines 526-7.

Figure 4H-K: Here the authors show in a complex in vitro series of assays that supernatants of Afa treated HaCat cells induce activation of dFB and their supernatant then induces monocyte migration. These results support the main hypothesis of the authors. However, an important information is missing about how this experiment has been performed. Have the supernatants been transferred directly or diluted? Would the supernatant of HaCat cells be able to induce migration directly?

Figure 5: In this figure the authors show that pre-treatment with HFD can rescue rats from the weight loss phenotype and partially rescue from the rash phenotype of Afa induced dWAT loss. The authors show here that this also results in changes in the immune cell composition in blood. To confirm the hypothesis of the authors, analysis of the immune cell infiltrate of the skin should be performed by FACS and IHC. This also accounts for Figure S6. In addition to cytokine receptors a cytokine and chemokine profile would give more information about rash severity.

The labelling of Figure 5 in the text and figure legend is not in correctly assigned

Figure 6: Here the authors show that prophylactic treatment of rats with Rosi can partially rescue from skin rash. These data are very promising. However, a more specific analysis of the parameters that have been shown before to be relevant for the skin rash such as IL-6 or monocyte infiltrate is missing. A therapeutic rather than prophylactic approach would also be an interesting and clinically relevant addition to this figure.

Figure S8: To show whether changes in rash severity would correspond to alterations in the anti-tumor response the authors treated nude mice bearing s.c. tumors with Afa or Afa + Rosi, could however detect no difference. However, the authors do not show, whether there was any induction of rash in this model, they only show weight loss. It has been published that EGFRi does not induce rash in adult mice. Therefore a similar experiment should be performed in rats.

---

## [Author Response]

Essential revisions:Figure 1: A timeline of the development of rash after Afa treatment is missing at the beginning. What is the percentage of rats developing rash grade 1, 2 or 3? Are there any other side effects of Afa treatment (diarrhea, lethality)?

We thank the reviewer for pointing this out. In the revised manuscript, we added the development process and rash grade percentages in Figure 1—figure supplement 1C based on seven experiments. The total number of rats was 39. According to our data, rats began to develop rash at day 3, while 28.21% of rats had grade 1 rash at day 4. Until day 7, almost all rats developed rash, 23.08% of which were grade 2. Grade 3 rash started to come out at day 8 and became more prevalent in the next two days.

Diarrhea is very common in Afa-treated rats (100%), corresponding to the high incidence rate in clinical patients (70%-95%) (Park et al., 2016; Sequist et al., 2013; Soria et al., 2015; Wu et al., 2014). According to our statistical data, the incidence rate of lethality was only 2.56% (1 out of 39).

Figure 1: The authors need to better describe their skin inflammation model. Is the rash hair follicle bound? What is the composition and location of the immune cell infiltrate and which cytokines and chemokines are dominant in the skin? Does the rash occur also with an specific EGFR inhibitor instead of the broad ErbB family inhibitor Afa (e.g. Erlotinib). Moreover, does the rash develop only in shaved areas or also in unshaved areas of the back skin? This should be clarified as a recent study associated EGFRi-mediated skin rash development with hair regrowth (Klufa et.)

We appreciate the reviewer for suggesting this important issue. We have conducted additional experiments to better describe the skin rash model, especially the hair eruption effect.

In the revised manuscript, we provided IHC results of several immune cells reported playing important roles in EGFRi-induced skin inflammation, including macrophage, mast cell, monocyte, and T cell. Consistent with the phenotypes in human skin biopsies and EGFR knockout mice models (Lichtenberger et al., 2013; Mascia et al., 2013), the rats’ skin showed enrichment of multiple inflammatory cells in the dermis (Figure 1—figure supplement 1E, F). The cytokine expression file also showed a similar tendency with human biopsies and mouse models (Lichtenberger et al., 2013; Mascia et al., 2013). As shown in Figure 1—figure supplement 1G, Afa dramatically increased the expression of proinflammatory chemokines such as *Ccl2*, *Ccl5*, *Cxcl1*, *Cxcl2*, and *Cxcl3*.

The rash can also be induced by many specific Erbb1 inhibitors. All three generations of EGFR inhibitors in the clinic have very high incidence rates of cutaneous toxicity (Supplementary file 1). In the revised version, we provided rash models induced by both first-generation EGFRi, Erlotinib, Gefitinib, and the third-generation EGFRi, Osimertinib. As shown in Figure 1—figure supplement 1D, the rash caused by Erlotinib, Gefitinib, and Osimertinib had the same phenotypes with Afatinib-induced rash.

In a recent study focusing on the effect of hair eruption on rash occurrence (Klufa et al., 2019), hair eruption-caused barrier defects have an important role in inducing rash phenotypes. Using waxing and tape stripping to remove the hair of mice could immediately induce de novo hair eruption. Mice with hair-removing had abnormal phenotypes of rash, including hair growth retardation and increased cytokine expression, whereas normal adult mice couldn’t show any barrier defects and inflammation during Erlotinib treatment. We conducted additional experiments (Figure 1—figure supplement 2). In our rat model, there was a similar situation. Shaving also had a significant impact on rash progress that the skin rash occurred earlier and the skin rash was more severe on shaved back skin than on the unshaved area (Figure 2—figure supplement 2). Nevertheless, the unshaved area still developed abnormal erythema (Figure 1—figure supplement 2A) upon the Afatinib administration. These results show that the rat model shares the similar symptoms and disease development time course with rash in humans and other rodent models, which provided a feasible model to study EGFRi-induced cutaneous toxicity.

Figure 1G: The authors show a correlation between rash grade and dWAT thickness. Is there a correlation to body weight? Rats show reduced food intake after Afa treatment (Suppl. Figure 1B). Could this be the cause for the reduction in dWAT? How does dWAT in rats react to fasting?Does shaving have any effect on Afa induced dWAT reduction? Are similar changes observed in unshaved skin?

Accumulation of dWAT paralleled with the increase of body weight in response to high-fat feeding (Kasza et al., 2016). However, there is no research focusing on the effect of fasting/weight loss treatment on dWAT. One research studied the effect of fasting on perigonadal adipose tissue found that fasting-induced lipolysis and decreased the adipocyte size (Kosteli et al., 2010). Decreased body weight and appetite could accompany with EGFRi therapy (supplementary file 1) (Park et al., 2016; Sequist et al., 2013; Soria et al., 2015; Wu et al., 2014). It could be a systematic effect caused by EGFRi or a result of multiple side effects including rash, diarrhea, vomiting, and stomatitis, etc. In addition, epidermal knockout EGFR mice showed a congenital dWAT reduction (Mascia et al., 2013). Thus, we think it is reasonable to hypothesize that body weight/food intake change had effects on dWAT, but it is inadequate to establish a direct and clear connection in the setting of EGFRi treatment.

There is evidence that dWAT plays an important role in maintaining thermal homeostasis. Mice housed at 31°C for 5 days displaced thinner dWAT layers compared to mice housed at constant temperature (21°C), while the cold stress imposed by housing at 4°C did not increase the dWAT layer (Kasza et al., 2014). In the revised manuscript, we provided additional experiments. We conducted histopathological examinations to investigate the effect of shaving on dWAT. In the unshaved areas, it seemed like their dWAT was a bit thinner at the initial time but without significant difference compared to the shaved areas (Figure 1—figure supplement 2C, D). This result might be due to the shaving-induced cold stress. In addition, the cold stress simultaneously began with the Afa treatment. We noted that dWAT in the unshaved areas also showed a similar decrease trend under long-term Afa treatment (Figure 1—figure supplement 2C, D).

Figure 1 E,F, Suppl. Figure 1C,-R: In these figures the labelling is unclear. The authors are labelling the columns Grade 0-3. Have all these animals been harvested on day 10 as mentioned in the methods section? Does grade 0 mean "before" Afa? What do the "control" samples then correspond to, since they also are assigned to rash grades?

We thank the reviewer for pointing out the labelling issue. To make it clearer, we supplemented a treatment scheme (Figure 1—figure supplement 1H). All animals received Afa treatment on Day 1, three rats were harvested at each rash grade time point. The time points of different rash grades were confirmed by statistical data from seven separate experiments (Figure 1—figure supplement 1C). In this figure, rats in grade 0 mean the control rats, they were harvested at day 0 without Afa treatment.

Figure 1H-J: PPARγ is strongly downregulated in the skin of Afa treated rats, but the KEGG Pparg signaling pathway is among the top 10 enriched pathways. How do the authors explain this discrepancy? It would be helpful to show all gene changes of the pathway and verify some gene changes on the protein level (e.g.: IHC). Where are these data from? No explanation or citation to a previous study can be found in the manuscript.

The KEGG enrichment represents the most significant changed 10 pathways. It doesn’t distinguish whether one pathway is upregulated or downregulated. To support the transcriptome data, we also conducted IHC staining of two important genes in PPAR signaling. The results showed that the protein level of PPARγ and Perilipin-2 agreed with the gene changes (Figure 1—figure supplement 4). The transcriptome data is from a previous study which is still under review, so it cannot be cited yet. Thus, we decided to remove Figure 1I and 1J.

Figure 2C and D: How do the authors take samples one day before the rash during the 10 day Afa treatment? How long have the rats been treated with Afa at the 1 day after rash timepoint?

We thank the reviewer for pointing out the lack of experimental details. We added the method at line 614. We harvested the rats according to our statistical data (Figure 1—figure supplement 1C). The mean time for rash occurrence is day 4, so we sacrificed rats one day before rash at day 3, and one day after rash at day 5 (Figure 1—figure supplement 1H). All rats received a 10-day treatment except rats that have been sacrificed previously.

Figure 2E: Authors should show the full timeline of the pictures (also the controls) and provide a quantification of adipocytes from each timepoint.

We thank the reviewer for this advice. In the revised manuscript, we provided the full timeline for all three groups (Ctrl, Afa, Afa+Rosi) and the quantification of both attached and dedifferentiated cells (Figure 2E to 2G).

Figure 3A: These experiments need to be quantified in addition to the representative pictures.

We appreciate the reviewer for reminding us about the quantification. In the revised version, we supplemented the area quantification of oil red (Figure 3B).

Figure 3C: How often has this experiment been repeated? Some changes are difficult to see. All the Western Blots throughout the manuscript should be quantified to be able to interpret these results: e.g. the claimed reduction of pAkt is not really visible and is also seen in the controls at day 9.

We thank the reviewer for pointing this out. We repeated this experiment five to eight times, replaced the previous data with obvious blots, and quantified the western blots. The inhibition and activation effects of Afa and Rosi on pAKT1 and pAKT2 were not synchronal with each other (Figure 3D and 3E).

Figure 3D: The population of adipocyte progenitors is not very well demarcated. It has been shown that among the CD34+CD29+ cells in the skin only a subpopulation is adipogenic, and this proportion is also depending on the hair cycle (Festa et al.,). The authors should stain also for CD24 and/or Sca1 to more precisely specify the proportion of adipogenic cells. The same accounts for Figure S7 B, D. (Figure S7D: it does not make sense to depict cells that are not there as MFI = 0):

We agree with the reviewer that among CD34+CD29+ cells there is only a subpopulation that is adipogenic. But there are no applicable FACS antibodies of Sca1 and CD24 for rats, and no research about dWAT has been reported on rats yet. However, there was a research using CD34+CD45-CD31- profile to identify the adipocyte precursor cells from retroperitoneal AT of rats (Spinedi and Giovambattista, 2016). Therefore, we had to use the CD31- CD45- CD34+ CD29+ subgroup to roughly evaluate the APs. For MFI evaluation, we thank the reviewer for pointing out that MFI=0 is not appropriate to describe cells without any fluorescence signals, so we deleted this figure.

Figure 3F: The authors claim that *S. aureus* outgrowth is because of impaired adipogenesis (text line 418). This is however an overstatement since a large number of cells (especially keratinocytes) in the skin have been shown to be able to produce antimicrobial factors that might cause this result. S. aureus outgrowth is a hallmark of EGFR-I induced skin inflammation and atopic dermatitis in general. The authors have no mechanistic claim that adipocytes are influencing the bacterial superinfection in this setting. The results are merely a correlation and a comparison of rodents with skin inflammation and without.

We appreciate the reviewer for pointing this important point out. We tried to establish the relationship between *S. aureus* infection and EGFRi-induced rash based on a well-accepted study from Lingjuan Zhang (Zhang et al., 2015). They reported that adipose precursor cells secret antimicrobial peptide cathelicidin during differentiation to against *S. aureus* infection. Mice with impaired adipogenesis were more susceptible to *S. aureus* infection. This conclusion gave us some insights into the relationship between *S. aureus* infection and EGFRi-induced skin inflammation. Unfortunately, the anti-CAMP antibody was made by the author’s lab and there are no mature products that can recognize CAMP in rats. To provide more mechanistic evidences, we conducted qPCR experiments to study the transcriptional level of the *Camp* gene both in dWAT and dFB cells isolated from rat skin (Figure 3I and 3J). dWAT in Afa group showed a lower expression level of *Camp* compared with control group. In addition, in different differentiation stages of dFB in vitro, transcriptional levels of *Camp* were decreased by Afa treatment while increased by Rosi. In summary, the data we collected depending on the currently available technologies could justify the causal relationship between EGFRi-induced dWAT reduction and *S. aureus* infection to some extent. However, we agree with the reviewer that this conclusion needs more direct evidence. Thus, in the revised manuscript, we have edited our writing to make the statement not that strong (line 281).

Figure 3G: Same as with Figure 3F: This represents a correlation with the general skin inflammation rather than a mechanistic explanation involving the adipocytes. Both statements (Figure 3G and H) should be ideally checked using a adipocyte specific cre line in a mouse model. Furthermore EGFR deletion in keratinocytes alone using K5-cre was enough to induce a dramatic *S. aureus* mediated dysbiosis and hair loss in mice (Klufa et al.,). The loss of adipocytes might still contribute to these effects but are not necessary.

We agree with the reviewer’s comment that adipocyte-specific knockout mouse model will be more appropriate to study the role of adipocytes. The results and conclusions of *S. aureus* infection and hair follicles were mostly based on previous research. These results need more specific and direct experiments to verify the causality with EGFR inhibition. We are seeking the opportunity for cooperation to build a dermal adipocyte knockout mouse model platform and hope to investigate the specific roles of dermal adipocytes in the future.

As for the epidermal EGFR deleted mice models, they represented similar cutaneous phenotypes with EGFRi-induced skin toxicity. There were hair follicles growth retardation, neutrophilic pustules, keratin plugs, microbiome outgrowth, etc. But the disadvantages of this mice model are the immense abnormal symptoms and high lethality. Most EGFR^△ep^ mutant mice died shortly after birth, and only less than 10% lived longer than 3 weeks (Lichtenberger et al., 2013). In addition, from the histopathological examination, we noticed that KO mice also showed a decrease of dWAT layer (Mascia et al., 2013). Combined with our finding that the inflammatory epidermis can promote lipolysis of dWAT. Therefore, we think that skin disease caused by EGFRi is a multifactorial condition, both keratinocytes and adipocytes contribute to its process, and maybe a mutual connection between these two cell types also plays an important role.

Figure 3H: here EGFR, pEGFR and ERK staining of adipocytes and their respective reduction after Afa treatment should also be shown.

We thank the reviewer for pointing this out. In the revised manuscript, we provided the pEGFR and pERK staining of adipocytes in Figure 3—figure supplement 1C.

Figure 3I: the quantification is missing. how many HFs have been counted to be able to make this statement?

We supplemented the quantification for apoptosis and proliferation of HFs (Figure 3—figure supplement 1D, E), *n*=6.

Figure S4C: In the text (line 444) the authors claim that there is a correlation between high levels of C18 FAs and increased lipolysis without showing any difference in the levels of this FA. This statement should be corrected. The same accounts for the statement in lines 526-7.

We agree with the reviewer that more evidences are needed to prove the correlation. We performed a mass spectrometry analysis of skin tissues from Ctrl and Afa groups after 3-day treatment to confirm the releasing of C18 FFAs. The result showed an increased tendency of C18:2 and other FFAs in the Afa group (Author response image 1). However, this increase had no significant statistic difference. This might due to the interference of sebaceous gland and dermal adipocytes. Therefore, we also adjusted the descriptions in the revised manuscript to make this statement not that strong (line 395).

**Author response image 1. sa2fig1:** C18 concentrations in skin tissues from Ctrl and Afa groups after 3-day treatment. *n*=3.

Figure 4H-K: Here the authors show in a complex in vitro series of assays that supernatants of Afa treated HaCat cells induce activation of dFB and their supernatant then induces monocyte migration. These results support the main hypothesis of the authors. However, an important information is missing about how this experiment has been performed. Have the supernatants been transferred directly or diluted? Would the supernatant of HaCat cells be able to induce migration directly?

Thanks for pointing this out. In the revised submission, we added more details of this experiment (Figure 4 legend, lines 324-326). The supernatants were transferred directly to the dFB-cultured wells. To ensure the nutrients for cell growth, we replaced a new culture medium of HaCaT cells before adding drugs. After 24 h treatment, the medium was transferred to dFB for another 24 h incubation. Thus, the total time for this culture medium was 48 h. In addition, the media used for HaCaT and dFB cell culture were both DMEM/high glucose, so it could be directly transferred. The supernatant of HaCat cells was able to induce migration directly. We performed the migration experiment treated directly by supernatants of Afa-treated HaCaT cells, it increased the monocyte migration. Nevertheless, we also used the undifferentiated-dFB to make a comparison with differentiated-dFB, the undifferentiated-dFB could approximately represent the effect of HaCaT alone. Compared with undifferentiated dFBs, the induction of migration was remarkable in differentiated dFBs (Figure 4O and 4P).

Figure 5: In this figure the authors show that pre-treatment with HFD can rescue rats from the weight loss phenotype and partially rescue from the rash phenotype of Afa induced dWAT loss. The authors show here that this also results in changes in the immune cell composition in blood. To confirm the hypothesis of the authors, analysis of the immune cell infiltrate of the skin should be performed by FACS and IHC. This also accounts for Figure S6. In addition to cytokine receptors a cytokine and chemokine profile would give more information about rash severity.

For the revised manuscript, we performed additional immune cell and cytokine IHC for both HFD and DCA experiments. The results have been incorporated into the revised manuscript. As shown in Figure 5—figure supplement 1C, the infiltration of macrophages, mast cells, monocytes, and T cells was decreased in HFD-fed rats. The expression of some inflammatory cytokines, such as IL6 and CCL2 showed a lower level in HFD-treated rats (Figure 5—figure supplement 1E). In the DCA experiment, infiltration of macrophages and monocytes showed a significant increase in DCA group, while the difference of T cells and mastocytes between vehicle and DCA was not very obvious (Figure 5—figure supplement 2J).

The labelling of Figure 5 in the text and figure legend is not in correctly assigned

We thank the reviewer for pointing this out. In the revised manuscript, we corrected this kind of mistakes.

Figure 6: Here the authors show that prophylactic treatment of rats with Rosi can partially rescue from skin rash. These data are very promising. However, a more specific analysis of the parameters that have been shown before to be relevant for the skin rash such as IL-6 or monocyte infiltrate is missing. A therapeutic rather than prophylactic approach would also be an interesting and clinically relevant addition to this figure.

We thank the reviewer for the advice. We conducted additional experiments, including the T cell infiltration, mastocyte cell infiltration, and IL6 infiltration. As shown in Figure 6—figure supplement 1G, Rosi could significantly reduce the inflammatory infiltration. We also performed a transcriptional experiment of inflammatory cytokines, Rosi showed obvious transcription inhibition on these genes (Figure 6—figure supplement 1F).

We agree with the reviewer that the therapeutic treatment has very promising application potential in the clinic. Therefore, we performed a therapeutic experiment in the revised version. After rats developed grade 1 rash, Rosi or Vehicle gel began to apply to the back skin of the rats. After nine-day treatment, rats that received Rosi had lower grade rash and higher body weight than the Vehicle group (Figure 6—figure supplement 2).

Figure S8: To show whether changes in rash severity would correspond to alterations in the anti-tumor response the authors treated nude mice bearing s.c. tumors with Afa or Afa + Rosi, could however detect no difference. However, the authors do not show, whether there was any induction of rash in this model, they only show weight loss. It has been published that EGFRi does not induce rash in adult mice. Therefore a similar experiment should be performed in rats.

The tumor study was performed in immunodeficient hosts and the immune-mediated process is important to induce cutaneous toxicity. Thus, the effect of Rosi on blocking rash could not be assessed. In addition, there are no suitable immune-deficient rat species, and normal rat species need rat-derived tumor cells to develop tumors, which is also very challenging to access. Considering the purpose of this study is to directly investigate the effect on the tumor, we applied Afa and Rosi treatment on nude mice.

References

Kasza I, Hernando D, Roldán-Alzate A, Alexander CM, Reeder SB. 2016. Thermogenic profiling using magnetic resonance imaging of dermal and other adipose tissues. *JCI Insight* 1:0–13. doi:10.1172/jci.insight.87146

Kasza I, Suh Y, Wollny D, Clark RJ, Roopra A, Colman RJ, MacDougald OA, Shedd TA, Nelson DW, Yen MI, Yen CLE, Alexander CM. 2014. Syndecan-1 Is Required to Maintain Intradermal Fat and Prevent Cold Stress. *PLoS Genet* 10. doi:10.1371/journal.pgen.1004514

Klufa J, Bauer T, Hanson B, Herbold C, Starkl P, Lichtenberger B, Srutkova D, Schulz D, Vujic I, Mohr T, Rappersberger K, Bodenmiller B, Kozakova H, Knapp S, Loy A, Sibilia M. 2019. Hair eruption initiates and commensal skin microbiota aggravate adverse events of anti-EGFR therapy. *Sci Transl Med* 11:1–18. doi:10.1126/scitranslmed.aax2693

Kosteli A, Sugaru E, Haemmerle G, Martin JF, Lei J, Zechner R, Ferrante AW. 2010. Weight loss and lipolysis promote a dynamic immune response in murine adipose tissue. *J Clin Invest* 120:3466–3479. doi:10.1172/JCI42845

Lichtenberger BM, Gerber PA, Holcmann M, Buhren BA, Amberg N, Smolle V, Schrumpf H, Boelke E, Ansari P, Mackenzie C, Wollenberg A, Kislat A, Fischer JW, Röck K, Harder J, Schröder JM, Homey B, Sibilia M. 2013. Epidermal EGFR controls cutaneous host defense and prevents inflammation. *Sci Transl Med* 5. doi:10.1126/scitranslmed.3005886

Mascia F, Lam G, Keith C, Garber C, Steinberg SM, Kohn E, Yuspa SH. 2013. Genetic ablation of epidermal EGFR reveals the dynamic origin of adverse effects of anti-EGFR therapy. *Sci Transl Med* 5. doi:10.1126/scitranslmed.3005773

Park K, Tan EH, O’Byrne K, Zhang L, Boyer M, Mok T, Hirsh V, Yang JCH, Lee KH, Lu S, Shi Y, Kim SW, Laskin J, Kim DW, Arvis CD, Kölbeck K, Laurie SA, Tsai CM, Shahidi M, Kim M, Massey D, Zazulina V, Paz-Ares L. 2016. Afatinib versus gefitinib as first-line treatment of patients with EGFR mutation-positive non-small-cell lung cancer (LUX-Lung 7): A phase 2B, open-label, randomised controlled trial. *Lancet Oncol* 17:577–589. doi:10.1016/S1470-2045(16)30033-X

Sequist L V., Yang JCH, Yamamoto N, O’Byrne K, Hirsh V, Mok T, Geater SL, Orlov S, Tsai CM, Boyer M, Su WC, Bennouna J, Kato T, Gorbunova V, Lee KH, Shah R, Massey D, Zazulina V, Shahidi M, Schuler M. 2013. Phase III study of afatinib or cisplatin plus pemetrexed in patients with metastatic lung adenocarcinoma with EGFR mutations. *J Clin Oncol* 31:3327–3334. doi:10.1200/JCO.2012.44.2806

Soria JC, Felip E, Cobo M, Lu S, Syrigos K, Lee KH, Göker E, Georgoulias V, Li W, Isla D, Guclu SZ, Morabito A, Min YJ, Ardizzoni A, Gadgeel SM, Wang B, Chand VK, Goss GD. 2015. Afatinib versus erlotinib as second-line treatment of patients with advanced squamous cell carcinoma of the lung (LUX-Lung 8): An open-label randomised controlled phase 3 trial. *Lancet Oncol* 16:897–907. doi:10.1016/S1470-2045(15)00006-6

Spinedi E, Giovambattista A. 2016. Long-Term Fructose Intake Increases Adipogenic Adipocyte Precursor Cells. doi:10.3390/nu8040198

Wu YL, Zhou C, Hu CP, Feng J, Lu S, Huang Y, Li W, Hou M, Shi JH, Lee KY, Xu CR, Massey D, Kim M, Shi Y, Geater SL. 2014. Afatinib versus cisplatin plus gemcitabine for first-line treatment of Asian patients with advanced non-small-cell lung cancer harbouring EGFR mutations (LUX-Lung 6): An open-label, randomised phase 3 trial. *Lancet Oncol* 15:213–222. doi:10.1016/S1470-2045(13)70604-1

Zhang L, Guerrero-juarez CF, Hata T, Bapat SP, Ramos R, Plikus M V, Gallo RL. 2015. Dermal adipocytes protect against invasive *Staphylococcus aureus* skin infection. *Science (80- )* 347:67–72.